# Highly Cytotoxic Osmium(II) Compounds and Their Ruthenium(II) Analogues Targeting Ovarian Carcinoma Cell Lines and Evading Cisplatin Resistance Mechanisms

**DOI:** 10.3390/ijms23094976

**Published:** 2022-04-29

**Authors:** Jana Hildebrandt, Norman Häfner, Daniel Kritsch, Helmar Görls, Matthias Dürst, Ingo B. Runnebaum, Wolfgang Weigand

**Affiliations:** 1Institut für Anorganische und Analytische Chemie Friedrich-Schiller Universität Jena, Humboldtstraße 8, 07743 Jena, Germany; jana.hildebrandt@astrazeneca.com (J.H.); helmar.goerls@uni-jena.de (H.G.); 2Department of Gynecology, Jena University Hospital—Friedrich-Schiller University Jena, Am Klinikum 1, 07747 Jena, Germany; norman.haefner@med.uni-jena.de (N.H.); daniel_kritsch@web.de (D.K.); matthias.duerst@med.uni-jena.de (M.D.)

**Keywords:** metal based compounds, cancer treatment, platinum resistance, ovarian cancer, ruthenium(II), osmium(II)

## Abstract

(1) Background: Ruthenium and osmium complexes attract increasing interest as next generation anticancer drugs. Focusing on structure-activity-relationships of this class of compounds, we report on 17 different ruthenium(II) complexes and four promising osmium(II) analogues with cinnamic acid derivatives as O,S bidentate ligands. The aim of this study was to determine the anticancer activity and the ability to evade platin resistance mechanisms for these compounds. (2) Methods: Structural characterizations and stability determinations have been carried out with standard techniques, including NMR spectroscopy and X-ray crystallography. All complexes and single ligands have been tested for cytotoxic activity on two ovarian cancer cell lines (A2780, SKOV3) and their cisplatin-resistant isogenic cell cultures, a lung carcinoma cell line (A549) as well as selected compounds on three non-cancerous cell cultures in vitro. FACS analyses and histone γH2AX staining were carried out for cell cycle distribution and cell death or DNA damage analyses, respectively. (3) Results: IC50 values show promising results, specifically a high cancer selective cytotoxicity and evasion of resistance mechanisms for Ru(II) and Os(II) compounds. Histone γH2AX foci and FACS experiments validated the high cytotoxicity but revealed diminished DNA damage-inducing activity and an absence of cell cycle disturbance thus pointing to another mode of action. (4) Conclusion: Ru(II) and Os(II) compounds with O,S-bidentate ligands show high cytotoxicity without strong effects on DNA damage and cell cycle, and this seems to be the basis to circumvent resistance mechanisms and for the high cancer cell specificity.

## 1. Introduction

### 1.1. Cisplatin and Analogues

The development of metals as anticancer agents began with the coincidental discovery of the biologic activity of *cis*-[Pt(NH_3_)_2_Cl_2_], Cisplatin by Rosenberg in 1965 [1]. Cisplatin was clinically approved in 1978 and targets primarily the DNA leading to DNA adducts, DNA damage, and apoptosis induction [2,3]. Nowadays, platin compounds are used in clinical anticancer treatment against cervical, bladder, head, and neck cancers as single agent and in combination therapy against testicular, ovarian, bladder, and head and neck cancers [4]. Unfortunately, the chemotherapy is limited by side effects, e.g., nephrotoxicity, ototoxicity, neurotoxicity, and innate and acquired resistant mechanism, which limit its clinical potencies [5,6]. Since 1992, the second-generation drug Carboplatin is approved worldwide, showing less nephro- and neurotoxicity than Cisplatin [3,4]. These drawbacks are the driving force for designing new drug candidates to improve the clinical efficacy of untargeted anticancer treatments [7,8,9,10].

### 1.2. Ruthenium Compounds for Anticancer Treatment

The development of potential ruthenium anticancer molecules started almost at the same time as the discovery of Cisplatin. Already 34 years before the discovery of Cisplatins’ potential, two researchers found the activity of Cs_2_[RuCl_6_]hydrate, a ruthenium(IV) species which showed inhibition of tumor growth [11]. Rosenberg himself discovered the activity of [Ru(NH_3_)Cl(OH)]Cl, a ruthenium(III) species [1,12]. The first ruthenium compounds were designed to mimic the platinum drugs and therefore had also am(m)ine and chlorido ligands, but more recent research showed that ruthenium based compounds have a different mode of action [4]. Additionally, ruthenium compounds are discussed as candidates for functionalized cancer-targeting drugs [13].

Clarke and coworkers introduced the ‘activation-by-reduction’-hypothesis, which is well accepted nowadays, implying that the ruthenium(III) drugs act as prodrugs that are reduced to their active species, ruthenium(II) [14]. The most promising candidates already analysed in clinical trials are tetrachloridobis(indazole)ruthenium(III), known as KP1019, NKP-1339 or IT-139 and tetrachlorido(dimethylsulfoxide)(imidazole)ruthenium(III), known as NAMI or NAMI-A [15,16,17]. KP1019, as well as IT-139 show fast binding to serum proteins in blood such as transferrin and albumin, which may regulate the tumor-specific activity of these compounds [16,18,19,20]. KP1019 induces apoptosis via the mitochondrial pathway and has completed Phase-1 clinical studies [21,22,23]. A change of the counter cation led to IT-139 which showed increased solubility allowing the application of higher drug concentrations and is presently the only compound undergoing Phase I/II-clinical studies [4,15,24]. Beside compounds with N-donor ligands, investigations and optimisations of S-donor ligands resulted in *trans*-[RuCl_4_(DMSO)(HIm)], whereas HIm is imidazole, known as NAMI (=Novel Anti-Tumor Metastasis Inhibitor) [4,17,25]. NAMI-A was the first ruthenium-based compound, which entered clinical trials and showed a selective activity against metastatic cells in vivo, but due to its poor clinical responses, clinical trials were interrupted [26,27,28].

Besides Ru(III) compounds, Ru(II) were analyzed for their biological activity. It is known that ruthenium(II) compounds are activated by a ligand exchange mechanism, especially by hydrolysis of the Ru-Cl bond [4,29]. Ruthenium(II) complexes, which are investigated for anticancer activity, show in general a typical ‘piano-stool’ geometry, with an η^6^-arene and three open coordination sites X, Y, Z for different ligands, which can lead to a charge of the complex itself. The arene ligand can be substituted (e.g., cymene), whereas Z is usually a halide. The positions X and Y can be two different monodentate ligands, but more common are bidentate ligands (e.g., N,N; N,O; O,O; or O,S) [30]. These organometallic ‘half-sandwich piano-stool’ compounds were investigated, mainly by the groups of Dyson, Sadler, and Keppler [31,32,33,34,35,36,37,38]. A great series of compounds, named RAPTA, were investigated by Dyson and coworkers showing antimetastatic properties, good aqueous solubility, as well as anti-angiogenic properties [31,32,39,40]. It is known that their primary target is not the DNA, as they show interactions with proteins [41]. In vivo and in vitro studies showed that the RAPTA compounds are not cytotoxic to normal cells, but active against some tumor cells [40,42]. A second series, first introduced by Sadler and coworkers, are the RAED compounds, e.g., RM175 showing a mechanism of action similar to Cisplatin by interaction with guanine [33,43]. Both compounds, RAPTA-C and RM175 are in advanced clinical studies due to good in vivo results [31,44,45]. 

Next to N,N-chelating substances, different chelating ligands, e.g.,: N,O; O,O; C,N; and S,N have been reported in the last years [43,46,47]. Ruthenium(II) compounds with O,S-chelating ligands have been introduced and investigated by Keppler and coworkers. By comparing O,O- and O,S-chelating ligands, they identified that the change from O,O to O,S ligands increases the solubility and stability and result in lower IC50 values [45,48,49]. In 2016, we have shown the increased biological activity of one ruthenium(II) complex with a cinnamic acid derivative as O,S-chelating ligand compared to their platinum(II) analogues and analyzed the interaction with proteins [50,51]. In the same year, Keppler and coworkers compared first time ruthenium(II) and osmium(II) analogues with O,S ligands, together with iridium(II) and rhenium(II) complexes [48]. They investigated the impact of the leaving group (imidazole vs. chlorido) and the change of the metal, resulting in good IC50 values in general. However, the best IC50 values were generated by the ligand itself, without any complexation to metals, being a great difference to our compounds showing 50- to 200-fold lower IC50 values after complexation to the ruthenium(II) [48,51].

### 1.3. Osmium Compounds for Anticancer Treatment

Significant results in the ruthenium drugs have enhanced the interest in osmium compounds to develop anticancer drugs [52,53]. Therefore, a discussion of osmium compounds cannot be separated from their ruthenium analogues, as the first compounds of this class have been analogues of well-known ruthenium complexes, e.g., RAPTA-C, RM175, NAMI-A and KP1019 [4,44,52,53,54,55,56,57,58,59,60]. The comparison of the osmium compounds to their ruthenium counterparts often results in different biological behavior, especially related to anticancer activity [4,52,53,54,61]. According to the HSAB-principle, osmium is a softer metal compared to ruthenium and therefore results in different coordination preferences to biomolecules. Moreover, it is known that the metal-ligand exchange mechanisms are slower for the osmium compounds compared to their ruthenium analogues [4,29,52,62,63]. Therefore, many osmium compounds, mostly representing half-sandwich complexes, have been investigated for their biological activity in vitro and partly in vivo [52,53,57,58,61,63,64,65,66,67,68,69]. Some osmium(II) compounds show similarities to Cisplatin and Carboplatin [52,70].

Several studies focused on comparing ruthenium(II) and their osmium(II) analogues, e.g., the study of Keppler and coworkers with the first comparison of O,S-chelating ligands to these metals, as mentioned before [48]. Recently, it was shown that both the specific cell line and the present ligands determine which metal complex has superior cytotoxicity and that targeting topoisomerase IIα contributes to the effect [71]. To point out some other examples, in 2018 Carcelli and coworkers compared ruthenium(II) and osmium(II) thiosemicarbazone (S,N-chelating) complexes [72]. The investigated compounds exhibited lower resistance factors than Cisplatin and the ruthenium(II), and osmium(II) analogues showed cytotoxic activity in the same range [72]. However, 2-phenylbenzothiazole (S,N-chelating) complexes with osmium(II) exhibited higher in vitro cytotoxicity than ruthenium(II) compounds [73]. Likely, some osmium(II)-p-cymene complexes functionalized with alkyl or perfluoroalkyl groups complexes showed better results than their ruthenium(II) analogues and are more selective to cancer cells [52,74]. Osmium(II) compounds with arene ligand and phosphane co-ligand tend to be more cancer specific but less active on platinum resistant cells than their ruthenium counterparts [75]. The further biological investigations were endorsed by the important statement, that ruthenium compounds which show good in vivo results (e.g., RAPTA-C) are compounds with low or even no cytotoxic behavior in vitro [21,24,27,28,52]. In general, these results led to the conclusion that the osmium complexes ‘tend to be slightly more cytotoxic than their ruthenium counterparts*’* [52]—but which metal complex is more cytotoxic in vitro and/or in vivo depends on the ligand system [52,61,64,68]. 

Ruthenium and osmium compounds were mainly investigated to mimic the mode of action of platinum-based complexes [4]. Although both metals are the most advanced non-platinum metallodrugs, the major challenge is still the discovery of their molecular targets [4]. Several investigations, ours included, showed that the biological behavior of these compounds is different to Cisplatin and that the DNA is not the primary target [4,38,50,76]. Both the nature of the ligands and the change of the metal (from ruthenium to osmium) results in different anticancer activity, biological activity in general, and may enable the specific targeting of cancer cells or photodynamic therapy and a catalytic activity [14,52,77,78,79]. Keppler and coworkers investigated some general structure-activity-relationships for osmium(II) and ruthenium(II) complexes; they concluded that the effect of the chosen metal and its anticancer activity is highly ligand-dependent [4]. Ruthenium(II) complexes are more active than their osmium(II) analogues with O,O-chelating ligand systems, whereas N,O/N,N/C,N and S,N osmium(II) compounds show better results [4,58,59,60,66,67,70,80,81,82]. As mentioned above, to the best of our knowledge, only the Keppler group analyzed an O,S-chelating system while focusing on different leaving groups and metal centers but did not analyze effects on platinum resistant cells [48,71]. In this work, we analyze different ruthenium(II) complexes and some of their osmium(II) counterparts with O,S-chelating ligands for anticancer properties. Next to investigating the influence of the metal-exchange, we focus on the structure-activity-relationships of different cinnamic acid derivatives as O,S-bidentate ligands. As clinically relevant models, both platinum-sensitive and –resistant epithelial ovarian cancer (EOC) cell lines were chosen for the in vitro comparison of the compounds’ cytotoxic effect. While EOC is, in the majority of cases, a platinum-sensitive disease, eventually the majority of patients will relapse and develop a platinum resistance. Platinum resistance is the main limitation for a long-lasting successful therapeutic effect, thus contributing to the low five-year survival rate of approximately 40% [83]. 

## 2. Results and Discussion

### 2.1. Synthesis 

The general structure of the ligand-system and the metal compounds analyzed in this work is given in Figure 1.

Cinnamic acid derivatives L1–L18 were synthesized according to published procedures as described in the Appendix A [51]. For ruthenium(II) and osmium(II) complexes, the corresponding β-Hydroxydithiocinnamic acid ester is deprotonated at the vinyloge acid function with 1 equiv. *t*-BuOK and afterwards given to a 0.5 equiv. [(η^6^-*p*-cymene)MCl_2_]_2_ (M = Ru or Os) suspension in THF (Figure 1). By adding the yellow ester solution to the M(II)-dimer, the color turns dark red and the reaction is stirred over night at room temperature, followed by acidic work up and column chromatography (THF/DCM). 

### 2.2. Characterization

All compounds were characterized by NMR spectroscopy, mass spectrometry, and elemental analysis (see Method section). Results for L13–L18 are in common with those for L1–L12, which were reported earlier (see Appendix A) [51]. The chemical shifts in ^1^H NMR and ^13^C{^1^H} NMR spectra show significant changes after complexation to the metal(II) center for both ligand systems, the O,S-chelating and the arene ligand. Specific changes in the NMR spectra have been already discussed previously for corresponding platinum(II) compounds and are in good agreement for the metal(II) compounds this work is dealing with [51]. Interestingly, the signals of the methine protons are shifted to high-field as a result of their complexation with ruthenium(II)/osmium(II), whereas a low-field shift of the corresponding signals for the platinum(II) complexes were observed, as shown in Figure 2. This is potentially caused by the better donor ability of the cymene ligand. A high-field shift for the ^13^C isotope of the -C=S-group was observed previously in the ^13^C{^1^H} NMR spectra of the platinum(II) compounds after complexation and can be confirmed for the ruthenium(II)/osmium(II) complexes as well (see Method section and Table 1). Synthesis for the metal complexes starts with the symmetrical bimetallic complex [(η^6^-*p*-cymene)MCl_2_]_2_ and aromatic signals of the cymene ligand are observed as two doublets, whereas the isopropyl groups resulted in one doublet. Nevertheless, the complexation to the O,S-chelating ligand leads to an unsymmetrical structure and results in chemically non-equivalent aromatic protons and carbons. Thus, four aromatic doublets for the cymene and two doublets for the isopropyl groups in the ^1^H NMR spectra, as well as four (instead of two) aromatic carbon signals and two (instead of one) signal for the isopropyl groups in the ^13^C{^1^H} NMR spectra are detectable. For the mass spectra in general, the molecular peak is not observable, only a [M-Cl]^+^ fragment, comparable to literature data [45], and a further fragmentation pathway as observed for the β-Hydroxydithiocinnamic acid derivatives itself.

### 2.3. Stability Determination

To investigate the behaviour of the ruthenium(II) complexes Ru1, Ru3, and Ru8 in solution, we analysed kinetic measurements via ^1^H NMR spectroscopy (every 1 h, one spectra). NMR signals and behaviour of the ruthenium and osmium compounds is similar, but osmium(II) compounds show a slower ligand exchange mechanism and a higher stability in general [4]. The stability determinations for the osmium(II) compounds using NMR spectroscopy show no structural changes (data for Os3 Appendix A). However, ruthenium(II) compounds exhibit a reduced stability. All ^1^H NMR spectra show that the ruthenium(II) molecules are not stable in dmso-d_6_ solution. Figure 3 shows the results for Ru1 at 37 °C in dmso-d_6_. The blue spectrum displays the first measurement at t = 0 h and the double-doublets of the cymene ligand changed quickly and already disappeared after 24 h (red spectra). The detailed data showing all of the ^1^H NMR spectra for 72 h prove that already after 5 h measurements, the signals for the cymene ligand change to a new signal, resulting in a high-field shift (Appendix A). Additional Figure 3 shows that signals of the aromatic region change and the methine proton is disappeared after 24 h (detailed analysis proves a loss after 7 h, Appendix A). The same measurements were done also with dmso at room temperature. Similar changes in the spectra occur at room temperature and new species are detectable (exemplified for Ru1 in the Appendix A). However, slower speciation processes in comparison to 37 °C measurements occur, which is exemplarily represented by the disappearance of the double-doublets of the cymene ligand after 29 h (rt, Appendix A) vs. 5 h (37 °C, Appendix A). As reported earlier, dmso molecules are able to bind to the ruthenium(II) center by losing the cymene ligand and changing the structure to an octahedral metal(II) coordination sphere [50]. Thus, an explanation for the new species can be the binding of dmso molecules to the ruthenium(II) center after loss of the cymene ligand representing the new species in the ^1^H NMR spectra. To support this hypothesis, the ruthenium(II) complexes were measured under same conditions (rt, 72 h) in CD_2_Cl_2_, and it was shown that the compounds are stable under these conditions in the other solvent (see Appendix A). In conclusion, it is shown that the analysed Ru(II) compounds are able to react with dmso at room temperature as well as at 37 °C, but not with dichlormethane.

However, we used freshly prepared stock solutions in dmso for each experiment and diluted these stock solutions within 1 h at RT to the final concentration in cell culture medium (final dmso concentration 0.5%). Therefore, the stability and the potential generation of speciation products in cell culture medium is more relevant but presently unknown. Earlier data show minor changes of the UV-VIS spectra in aqueous solutions pointing to an aquation (ligand exchange chloride to aqua) [50]. Even more important, incubations in protein solution (RNaseA) prove the interaction and binding to proteins [50]. Therefore, it is likely that Ru and eventually Os compounds undergo protein binding and speciation processes in biological systems. Although, the species causative for observed biological effects (see below) is unknown, these effects are attributable to the tested compounds.

### 2.4. Molecular Structures

Ruthenium(II) complexes Ru9, Ru13, and Ru14 as well as L14, L15, L17, and L18 were characterized by means of single crystal X-ray structure determination, whereas the molecular structures of Ru3, L1, L3, L4, L8, and L9 are already known [50,51]. Figure 4 shows the ruthenium(II) complex 14, whereas the molecular structures of Ru9, Ru13, and of the ligands are depicted in Appendix A. Results are in good agreement with the values reported earlier [50].

Table 2 displays specific bond length and angles for the presented Ru(II) compounds. The ruthenium(II) center shows a tetrahedral structure environment with L-Ru-L angles of around 90°. The bond lengths of ruthenium (here for example Ru9) and their neighbouring atoms are decreasing in the order of S(1)-Ru(1) (2.3544(5)) > Cl(1)-Ru(1) (2.4081(5)) > O(1)-Ru(1) (2.0790 (14) Å). The bond lengths of the oxygen-substituted moiety at the aromatic ring O(2)-C(9/8/7) are in the same range, whereas the bond lengths for *ortho*-substituted Ru9 (1.359(3) Å) are the smallest. Coordination of the O,S-chelating ligands to ruthenium(II) results in the elongation of the C(1)-S(1) bond and shortening of the C(3)-O(1) bond; this is comparable to the already discussed platinum(II) complexes [51].

### 2.5. Biological Behavior

The biological behaviour of all substances was characterized by their cytotoxic activity against a panel of cell lines enabling an understanding of the structure-activity relationship. Cytotoxic activity was determined on ovarian carcinoma cell lines SKOV3 and A2780 as well as their Cisplatin resistant analogues (SKOV3cis and A2780cis) [84,85] and the lung carcinoma cell line A549. Due to a low solubility in water, dmso is used as a solvent for the preparation of a dilution series in cell culture experiments. The toxic influence of dmso was determined earlier and experiments were carried out with 0.5% dmso in cell culture media and this concentration was used as reference sample in each MTT assay (details: Section 3) [51]. Cisplatin was used as a reference substance, and a 4.7 or 3.6 times higher IC50 value was observed for resistant cell lines; see Table 3. Resistance factors (RF) were determined for all substances (for IC50 values and RF of β-Hydroxydithiocinnamic acid esters L1–L18, see Appendix A). All investigated ruthenium(II) compounds show lower RF values than Cisplatin on ovarian carcinoma cell lines, ranging from 0.2 to 1.5 (Table 3). Whereas the IC50 values on the non-resistant cell lines are in most cases higher than the IC50 of the reference substance, no increase of IC50 values is observed for the resistant cell lines. Contrary, eight ruthenium complexes show lower IC50 values on SKOV3cis than Cisplatin and four compounds on A2780cis. Thus, it can be concluded that these compounds are able to bypass the Cisplatin resistance mechanism in these cell lines pointing to a different mechanism of action. 

The osmium compounds show, in most cases, lower IC50 values than the reference Cisplatin (except for SKOV3 and Os7, 13, and 14, Table 3). To point out, all substances show IC50 values between 0.3–0.4 μM on A2780, whereas Cisplatin has an IC50 value of 1.3 μM. On the resistant analogue of A2780, the activity is more than five times higher for Os3 (0.4 μM) and Os13 (0.8 μM) in comparison to Cisplatin (6.1 μM). Albeit only one compound (Os3) exhibits a lower IC50 value for SKOV3 than Cisplatin, all compounds have a higher activity against SKOV3cis. Remarkably, Os7 shows a 13-times lower IC50 value than Cisplatin (0.6 to 13.5 μM). The most promising candidate, Os3, shows IC50 values between 0.4 μM (A2780) and 2.3 μM (SKOV3cis), generally lower than the range of Cisplatin (1.3 μM A2780–13.5 μM SKOV3cis). Whereas the resistance factors of the ruthenium compounds are in most cases lower than 1, pointing to the specific targeting of resistant cells, the osmium analogues do not behave the same. This confirms earlier published comparison studies showing that osmium analogues of ruthenium complexes exhibit a different biological behavior in vitro (see introduction).

Table 4 shows IC50 values for normal primary short-term cell cultures of keratinocytes and fibroblasts as well as the non-cancerous breast epithelial cell line MCF10A. As mentioned before, Cisplatin exhibits numerous side effects by its unselective behaviour and cytotoxic activity against normal cells, which is also reflected by the measured IC50 values against non-cancerous cell cultures. Despite this, both most active compounds, Os3 and Ru14, show high IC50 values for these cells.

To conclude, the osmium compounds are in general more active against all five cell lines than Cisplatin and their ruthenium counterparts. This shows the enormous potential for osmium compounds as next generation anticancer drugs. However, the ruthenium compounds are specifically active against Cisplatin-resistant cell lines, meaning they are able to elude the mechanisms of Cisplatin resistance. This indicates the opportunity for ruthenium compounds to be selected for resistant tumors. Additionally, our data showing a higher activity for osmium compared to ruthenium compounds in ovarian and lung cancer cell lines resemble data from Klose et al., identifying tumor type specific activity ratios for isosteric Ru/Os compounds using the NCI-60 cell line panel [86]. Both compound classes do not attack non-cancerous cells resulting in higher cancer specificity compared to Cisplatin. This is confirmed by recent data for breast cancer cell lines showing a high cancer cell selectivity for similar Ru(II) complexes with cinnamic acid derivates [87] and for Ru(II)/Os(II) complexes with N,N-bidentate ligands in various cancer cell models [88]. This higher cancer cell specificity potentially leads to lower side effects during the therapy in vivo. Lower side effects may translate into the treatment with higher doses of the drugs, resulting in earlier and increased effects. Therefore, acquired drug resistance mechanisms arising after several treatments with suboptimal doses may be circumvented by drugs like the osmium compounds due to lower toxic side effects. Altogether, there are possibly two different indications for the ruthenium(II) and osmium(II) complexes. The ruthenium(II) compounds should be further developed for a treatment of Cisplatin resistant tumors, whereas the osmium(II) complexes can be an alternative for the first-line therapy due to higher cytotoxic activity compared to Cisplatin. 

A further analysis for the different ruthenium(II) compounds to determine structure-activity-relationships shows that five compounds (Ru14, Ru15, Ru2, Ru5, and Ru3) exhibit lower mean IC50 value on Cisplatin resistant cell lines than Cisplatin itself (Appendix A). Interestingly, compounds Ru14, Ru15, and Ru16 are, all together, the most active compounds in comparison to Cisplatin (Figure 5A). The compound Ru14 shows a lower mean IC50 value than Cisplatin for all cancer cell lines (Appendix A), for all ovarian carcinoma cell lines (Appendix A), for the Cisplatin resistant cell lines (Appendix A), and for all non-resistant cell lines (Appendix A). In conclusion, the determined structure-activity-relationship shows that longer alkyl chains at the aromatic ring lead to higher cytotoxic activity. The most active compound having an ethoxy-group at *para*-position (Ru14) is followed by Ru15 with an ethoxy-group at *ortho*-position. Interestingly, compound Ru16 has a butoxy-substituent at *meta*-position. Thus, it can be concluded that the biological activity is mediated by a longer chain (butoxy) at the *meta*-position, whereas the *ortho*- and *para*-positions are more suitable with a shorter chain (ethoxy). To have a further look at the influence of the different ligand systems and substitution patterns, all β-Hydroxydithiocinnamic acid alkyl esters were tested under same conditions as their derived ruthenium(II) complexes (Figure 5B, Appendix A). Figure 5B shows the trend of all IC50 values ordered by an increased mean IC50 value (determined for all five cell lines) for the β-Hydroxydithiocinnamic acid alkyl esters. Interestingly, the most active compounds are L17, L14, L18, L16, L13, and L15, showing similar low IC50 values on all cell lines. This confirms the results for the corresponding ruthenium(II) complexes, proving that the longer alkyl chains on the aromatic positions are the most active compounds and that the IC50 values increase by decreasing lipophilicity. All ligands are less cytotoxic than Cisplatin itself and therefore the metal(II) center is necessary for the high cytotoxic activity, what is in clear contrast to the literature for O,S-chelating ruthenium(II) or osmium(II) compounds with thiomaltol ligand [48,71] but confirmed for Ru(II)/Os(II) compounds with N,N-bidentate glycosyl heterocyclic ligands [88]. This is exemplarily shown by the comparison of mean IC50 values for Cisplatin, L14, and Ru14 (Appendix A). The ruthenium(II) center strongly decreases the IC50 values in all cases, and therefore the metal is the active part that is supported by the most active ligand system. 

The reduced viability under treatment, as measured by the MTT assay, can be a result of cell cycle arrest and/or increased cell death. To further evaluate the anticancer properties of the ruthenium(II) complexes we measured cell cycle distribution and cell death rates after treatment with Ru3 or Ru14. After seeding and attaching, the cells were treated for 48 h with different concentrations of substances. For cell cycle distribution measurements, a recovery phase of 24 h was added after treatment, and cells were fixed and stained with PI for the DNA content. Arresting of cells in specific cell cycle phases gives them time to resolve the DNA damage (G1 arrest) or is an initial step to apoptosis if DNA damage is too severe (G2/M arrest) [89]. As previously shown, Cisplatin (5 µM) efficiently induces cell cycle arrest in G2/M phase in parental A2780 and SKOV3 cells, whereas resistant cells show only a minor G2/M arrest [85]. On the other side, both examined ruthenium complexes show no or only a minor effect on cell cycle distribution (Figure 6). This is in line with other published ruthenium(II) complexes, which do not all induce cell cycle arrest [89,90]. Therefore, one can suggest that these complexes do not induce high DNA damage levels, leading to cell cycle arrest.

For cell death rate analysis, live cells were stained with PI immediately after 48 h treatment. Again, it can be seen that 15 µM Cisplatin efficiently induces cell death in parental ovarian cancer cells [85], where it is 29.9-fold higher for A2780 and 6.3-fold higher for SKOV3 compared to untreated cells. Furthermore, resistant cells show much lower Cisplatin-induced cell death rates (Figure 7). Both complexes, Ru3 and Ru14, have a high capacity to induce cell death in vitro (Figure 7). In A2780 cells, both compounds trigger similar cell death rates in parental and Cisplatin-resistant cells. Cisplatin-resistant SKOV3 are much more sensitive to both ruthenium(II) complexes than the parental counterpart, with a median of 3.3-fold higher sensitivity. Interestingly, Ru3 induced higher cell death rates than Ru14 despite contrary results for IC50 values.

Previous studies showed a direct induction of apoptosis by ruthenium(II) complexes via ROS production and activation of pro-apoptotic BCL2-family proteins [91,92]. Ru(II) compounds may also inhibit TrxR (thioredoxin reductase), thus resulting in ROS production, mitochondrial dysfunction, and apoptosis [93]. ROS production may also lead to endoplasmatic reticulum stress-induced apoptosis [36]. In general, many ruthenium(II) complexes with different ligands induce intracellular ROS [94,95,96,97,98,99]. Moreover, the cytotoxic activity of Os(II) compounds can be inhibited by vitamin E co-treatment pointing to the contribution of ROS [88]. In addition to mitochondrial dysfunction, Ru(II) complexes may affect glycolysis [100] or topoisomerase I/II, thus inducing necroptosis [101]. Ru(II) compounds with modified pyrithione ligands were recently described to overcome platinum resistance in ovarian cancer cells by inducing cytostatic G1 arrest, TrxR inhibition, and cell membrane damage [102]. Both Ru(II) and Os(II) compounds can also inhibit proteosynthesis [34,103]. A direct interaction of Ru3 with a model protein (RNaseA) resulted in ligand exchange, binding to histidine residues, and altered coordination sphere geometry, pointing to a mode-of-action that involves protein targets [50]. Future studies may identify the specific target proteins enabling molecular docking studies and specific refinement of the organo-metal compound structure. Altogether, presented compounds may use some of these alternative modes of action as well, as we see efficient cell death but no cell cycle arrest induction by Ru3/Ru14. Furthermore, the ruthenium(II) core atom might be responsible for this effect because of the lack of anticancer behaviour of the ligand L14 (Appendix A). To further confirm that Ru compounds use another mechanism of action, DNA damage analyses were conducted for Ru3 and Ru14 (Figure 8). Both Ru compounds (at IC50 concentration) induced less γH2AX-foci as Cisplatin after 24h incubation under the same conditions. This confirms published data pointing to a DNA-independent mode of action for ruthenium compounds [41,71,104]. However, other data show an interaction of ruthenium complexes with DNA [33,43,99,105,106]. These contrary observations may relate to experimental conditions or specific ligands.

The presented data suggest that Ru(II) and Os(II) complexes with O,S-chelating β-Hydroxydithiocinnamic acid esters are both highly active and specific against cancer cell lines (Os(II) compounds) or Cisplatin resistant cancer cells (Ru(II) compounds). The avoidance of resistance mechanisms seems to be related to another mode of action inducing cell death without high levels of DNA damage or cell cycle arrest. Several limitations must be discussed for the evaluation of these data. Firstly, the biological activity was determined for in-vitro 2D cell culture systems, only. Further analyses in 3D cell cultures or in vivo should clarify the potential for clinical use of the most active compounds (Os3, Ru14). Thereby, the detailed mode-of-action must be identified, although first data point to a potential contribution of protein interactions [50]. Secondly, presented and already published data point to the instability of Ru(II) compounds and the generation of speciation products in dmso and biological systems (Figure 3) [50]. Therefore, it is presently unknown which specific compound directly causes the observed biological effects. However, our experiments show clearly and reproducibly that the tested complexes are the general source of the effects. If future studies can solve these limitations and validate the high cancer cell specific cytotoxicity also against platin resistant tumors, these compounds are likely to improve the treatment of ovarian cancer patients.

## 3. Materials and Methods

### 3.1. Materials and Techniques

All reactions were performed using standard Schlenk and vacuum-line techniques under nitrogen atmosphere. The NMR spectra were recorded with a Bruker Avance 200 MHz, 400 MHz, or 600 MHz spectrometer. Chemical shifts are given in ppm with reference to SiMe_4_. Mass spectra were recorded with a Finnigan MAT SSQ 710 instrument. Elemental analysis was performed with a Leco CHNS-932 apparatus. Silica gel 60 (0.015–0.040 mm) was used for column chromatography, and TLC was performed using Merck TLC aluminium sheets (Silica gel 60 F_254_). Chemicals were purchased from Fisher Scientific (Schwerte, Germany), Sigma-Aldrich (Taufkirchen, Germany), or Acros (Nidderau, Germany) and were used without further purification. All solvents were dried and distilled prior to use according to standard methods.

### 3.2. Synthesis

Different β-Hydroxydithiocinnamic acid alkyl esters and [(η^6^-*p*-cymene)XCl_2_]_2_ (X = Ru or Os) were prepared by modified literature methods [51,107]. New compounds L13–L17 are described in the Appendix A.

General procedure 1: Ruthenium(II) complexes with β-Hydroxydithiocinnamic acid alkyl esters, chlorido and *p*-cymene as ligands (Ru1–Ru17).

[(η^6^-*p*-cymene)RuCl_2_]_2_ (0.5 equiv.) was dissolved in 50 mL tetrahydrofurane (THF). The corresponding ligand L1-L12 (1 equiv.) was solved in 25 mL THF and potassium-*tert*.-butoxylate (*t*-BuOK, 2 equiv.) was added to that solution and stirred 30 min at rt. The solution of the deprotonated ligand was added dropwise to the suspension of [(η^6^-*p*-cymene)RuCl_2_]_2_ and stirred at room temperature for 24 h. After adding sulfuric acid (H_2_SO_4_, 20 mL, 2M) to the solution, the mixture was stirred for 30 min at rt and afterwards extracted with dichlormethane (DCM, 3 × 30 mL). The combined organic phases were washed with water (3 × 20 mL), dried over sodium sulfate and after filtration and evaporation of the solvent the crude product was purified with column chromatography.

#### 3.2.1. General Procedure 1: Osmium(II) Complexes with β-Hydroxydithiocinnamic Acid Alkyl Esters, Chlorido, and p-cymene as Ligands (Os1–Os4)

[(η^6^-*p*-cymene)OsCl_2_]_2_ (0.5 equiv.) was dissolved in 50 mL tetrahydrofurane (THF). The corresponding ligand (1 equiv.) was solved in 25 mL THF, and potassium-*tert*.-butoxylate (*t*-BuOK, 2 equiv.) was added to that solution and stirred 30 min at rt. The solution of the deprotonated ligand was added dropwise to the suspension of [(η^6^-*p*-cymene)RuCl_2_]_2_ and stirred at room temperature for 24 h. After adding sulfuric acid (H_2_SO_4_, 20 mL, 2M) to the solution, the mixture was stirred for 30 min at rt and afterwards extracted with dichlormethane (DCM, 3 × 30 mL); the combined organic phases were washed with water (3 × 20 mL), dried over sodium sulfate and after filtration and evaporation of the solvent the crude product was purified with column chromatography.

#### 3.2.2. [(n^6^-p-cymene)Ru(1-phenyl-3-(methylthio)-3-thioxo-prop-1-en-1-olate-O,S)Cl] (Ru1)

Synthesis was performed according to general procedure 1. [(η^6^-*p-*cymene)RuCl_2_]_2_ (500 mg, 0.81 mmol) was used. L1 (341 mg, 1.62 mmol) was dissolved in THF, *t*-BuOK (182 mg, 1.62 mmol) was added. Column chromatography mobile phase: DCM-DCM 10:THF 1-THF. Yield: 540 mg (69.5%) as red crystals. ^1^H NMR (600 MHz, CD_2_Cl_2_): δ = 1.29 (dd, ^3^*J_H-H_* = 5.8 Hz, ^4^*J_H-H_* = 2.2 Hz, 6H, -cymene-CH-(C*H*_3_)_2_); 2.20 (s, 3H, -cymene-C*H*_3_); 2.69 (s, 3H, -SC*H*_3_); 2.85 (sp, 1H, -cymene-C*H*-(CH_3_)_2_); 5.29 (d, ^3^*J_H-H_* = 5.9 Hz, 2H, -cymene:CH_3_-C-C*H*-CH-C-CH-(CH_3_)_2_); 5.50 (dd, ^3^*J_H-H_* = 22.6 Hz, ^4^*J_H-H_* = 5.9 Hz, 2H –cymene:CH_3_-C-CH-C*H*-C-CH-(CH_3_)_2_); 6.76 (s, 1H, = C*H*); 7.40 (m, 2H, -Ar-*m*-H); 7.48 (m, 1H, -Ar-*p*-H); 7.81 (d, ^3^*J_H-H_* = 7.4 Hz, 2H, -Ar-*o*-H). ^13^C{^1^H} NMR (101 MHz, CD_2_Cl_2_): δ = 17.4 (-SCH_3_); 18.1 (-cymene-C-*C*H_3_); 22.3 (-cymene-CH-(*C*H_3_)_2_); 30.8 (-cymene-*C*H-(CH_3_)_2_); 83.1 (-cymene:CH_3_-C-*C*H-CH-C-CH-(CH_3_)_2_); 83.3 (CH_3_-C-*C*H-CH-C-CH-(CH_3_)_2_); 85.5 (CH_3_-C-CH-*C*H-C-CH-(CH_3_)_2_); 100.4 (CH_3_-*C*-CH-CH-C-CH-(CH_3_)_2_); 102.5 (CH_3_-C-CH-CH-*C*-CH-(CH_3_)_2_); 109.1 (=*C*H); 127.4 (-Ar-*o*-C); 131.3 (-Ar-*p*-C); 140.0 (-Ar-C1); 178.0 (-*C*-O-); 187.8 (-*C*=S). MS (DEI): m/z = 444, 438, 399, 394, 317, 315, 280, 274. Elemental analysis: calculated for C_20_H_23_ClORuS_2_ C: 50.04%; H: 4.83%, found: C: 49.92%; H: 4.82%.

#### 3.2.3. [(η^6^-p-cymene)Ru(1-phenyl-3-(ethylthio)-3-thioxo-prop-1-en-1-olate-O,S)Cl] (Ru2)

Synthesis was performed according to general procedure 1. [(η^6^-*p-*cymene)RuCl_2_]_2_ (500 mg, 0.81 mmol) was used. L2 (363 mg, 1.62 mmol) was dissolved in THF, *t*-BuOK (182 mg, 1.62 mmol) was added. Column chromatography mobile phase: DCM-DCM 10:THF 1-THF. Yield: 460 mg (57.3%) as red crystals.^1^H NMR (600 MHz, CD_2_Cl_2_): δ = 1.29 (d, ^3^*J_H-H_* = 7.0 Hz, 6H, -cymene-CH-(C*H*_3_)_2_); 1.40 (t, ^3^*J_H-H_* = 7.5 Hz, 3H, -S-CH_2_-C*H*_3_); 2.21 (s, 3H, -cymene-C*H*_3_); 2.85 (m, 1H, -cymene-C*H*-(CH_3_)_2_); 3.28 (q, ^3^*J_H-H_* = 7.5 Hz, 2H, -S-C*H*_2_-CH_3_); 5.28 (d, ^3^*J_H-H_* = 5.8 Hz, 2H, -cymene:CH_3_-C-C*H*-CH-C-CH-(CH_3_)_2_); 5.50 (d, ^3^*J_H-H_* = 5.28 Hz, 2H, –cymene:CH_3_-C-CH-C*H*-C-CH-(CH_3_)_2_); 6.75 (s, 1H, =C*H*); 7.40 (m, 2H, -Ar-*m*-H); 7.48 (m, 1H, -Ar-*p*-H); 7.80 (d, ^3^*J_H-H_* = 7.4 Hz, 2H, -Ar-*o*-H). ^13^C{^1^H} NMR (101 MHz, CD_2_Cl_2_): δ = 13.9 (-SCH_2_C*H*_3_); 18.2 (-cymene-C-*C*H_3_); 22.4 (-cymene-CH-(*C*H_3_)_2_); 24.2 (-S-*C*H_2_CH_3_); 30.9 (-cymene-*C*H-(CH_3_)_2_); 83.1 (-cymene:CH_3_-C-*C*H-CH-C-CH-(CH_3_)_2_); 83.3 (CH_3_-C-*C*H-CH-C-CH-(CH_3_)_2_); 85.6 (CH_3_-C-CH-*C*H-C-CH-(CH_3_)_2_); 85.6 (CH_3_-C-CH-*C*H-C-CH-(CH_3_)_2_); 100.6 (CH_3_-*C*-CH-CH-C-CH-(CH_3_)_2_); 102.6 (CH_3_-C-CH-CH-*C*-CH-(CH_3_)_2_); 109.4 (=*C*H); 127.5 (-Ar-*o*-C); 131.4 (-Ar-*p*-C); 140.1 (-Ar-C1); 178.0 (-*C*-O-); 187.1 (-*C*=S). MS (DEI): m/z = 458, 456, 399, 393, 311, 297. Elemental analysis: calculated for C_21_H_25_ClORuS_2_ C: 51.05%; H: 5.10%, found: C: 50.97%; H: 5.03%.

#### 3.2.4. [(η^6^-p-cymene)Ru(1-(3-hydroxyphenyl)-3-(methylthio)-3-thioxo-prop-1-en-1-olate-O,S)Cl] (Ru3)

Synthesis was performed according to general procedure 1. [(η^6^-*p*-cymene)RuCl_2_]_2_ (500 mg, 0.81 mmol) was used. L3 (367 mg, 1.62 mmol) was dissolved in THF, *t*-BuOK (182 mg, 1.62 mmol) was added. Column chromatography mobile phase: DCM-DCM 10:THF 1-THF. Yield: 190 mg (23.6%) as red crystals. ^1^H NMR (600 MHz, CD_2_Cl_2_): δ = 1.26 (d, ^3^*J_H-H_* = 6.4 Hz, 6H, -cymene-CH-(C*H*_3_)_2_); 2.20 (s, 3H, CH_3_, -cymene-C*H*_3_); 2.64 (s, 3H, -SC*H*_3_); 2.83 (m, 1H, -cymene-C*H*-(CH_3_)_2_); 5.33 (m, 2H, -cymene:CH_3_-C-C*H*-CH-C-CH-(CH_3_)_2_); 5.52 (m, 2H –cymene:CH_3_-C-CH-C*H*-C-CH-(CH_3_)_2_); 6.71 (s, 1H, =C*H*); 6.85 (m, 2H, -Ar-*o*-H); 7.11 (m, 1H, -Ar-*m*-H); 7.23 (m, 3H, =C*H*/-Ar-*p*-H); 10.1 (s, 1H, -CO*H*). ^13^C{^1^H} NMR (101 MHz, CD_2_Cl_2_): δ = 17.6 (-SCH_3_); 18.3 (-cymene-C-*C*H_3_); 22.4 (-cymene-CH-(*C*H_3_)_2_); 30.9 (-cymene-*C*H-(CH_3_)_2_); 83.3 (-cymene:CH_3_-C-*C*H-CH-C-CH-(CH_3_)_2_); 83.8 (CH_3_-C-*C*H-CH-C-CH-(CH_3_)_2_); 85.5 (CH_3_-C-CH-*C*H-C-CH-(CH_3_)_2_); 85.6 (CH_3_-C-CH-*C*H-C-CH-(CH_3_)_2_); 100.8 (CH_3_-*C*-CH-CH-C-CH-(CH_3_)_2_); 102.3 (CH_3_-C-CH-CH-*C*-CH-(CH_3_)_2_); 109.2 (=*C*H); 125.2 (-Ar-*m*-C); 129.2 (-Ar-*o*-C); 129.4 (-*C*OH); 156.9 (-Ar-*p*-C); 178.0 (-Ar*C*1); 187.3 (-*C*-O-); 207.2 (-*C*=S). MS (DEI): m/z = 134, 119, 115, 91, 77, 39, 28. Elemental analysis: calculated for C_20_H_23_ClO_2_RuS_2_ C: 48.43%; H: 4.67%, found: C: 48.60%; H: 4.83%.

#### 3.2.5. [(η^6^-p-cymene)Ru(1-(4-hydroxyphenyl)-3-(methylthio)-3-thioxo-prop-1-en-1-olate-O,S)Cl] (Ru4)

Synthesis was performed according to general procedure 1. [(η^6^-*p*-cymene)RuCl_2_]_2_ (500 mg, 0.81 mmol) was used. L4 (367 mg, 1.62 mmol) was dissolved in THF, *t*-BuOK (182 mg, 1.62 mmol) was added. Column chromatography mobile phase: DCM-DCM 6:THF 1-THF. Yield: 190 mg (23.6%) as red crystals. ^1^H NMR (400 MHz, CD_2_Cl_2_): δ = 1.20 (m, 6H, -cymene-CH-(C*H*_3_)_2_); 2.10 (s, 3H, -cymene-C*H*_3_); 2.59 (s, 3H, -SC*H*_3_); 2.75 (m, 1H, -cymene-C*H*-(CH_3_)_2_); 5.45 (d, ^3^*J_H-H_* = 5.7 Hz, 2H, -cymene:CH_3_-C-C*H*-CH-C-CH-(CH_3_)_2_); 5.67 (d, ^3^*J_H-H_* = 20.4 Hz, 2H –cymene:CH_3_-C-CH-C*H*-C-CH-(CH_3_)_2_); 6.65 (s, 1H, =C*H*); 6.80 (m, 2H, -Ar-*o*-H); 7.76 (m, 2H, -Ar-*m*-H): 10.1 (s, 1H, -CO*H*). ^13^C{^1^H} NMR (101 MHz, CD_2_Cl_2_): δ = 17.5 (-SCH_3_); 18.2 (-cymene-C-*C*H_3_); 22.2/22.6 (-cymene-CH-(*C*H_3_)_2_); 30.8 (-cymene-*C*H-(CH_3_)_2_); 82.9 (-cymene:CH_3_-C-*C*H-CH-C-CH-(CH_3_)_2_); 84.4 (CH_3_-C-*C*H-CH-C-CH-(CH_3_)_2_); 85.1 (CH_3_-C-C-*C*H-C-CH-(CH_3_)_2_); 85.5 (CH_3_-C-CH-*C*H-C-CH-(CH_3_)_2_); 100.6 (CH_3_-*C*-CH-CH-C-CH-(CH_3_)_2_); 102.2 (CH_3_-C-CH-CH-*C*-CH-(CH_3_)_2_); 108.5 (=*C*H); 125.2 (-Ar-*o*-C); 129.9 (-Ar-C1); 130.9 (-Ar-*m*-C); 160.7 (-*C*OH); 177.8 (-*C*-O-); 185.0 (-*C*=S). MS (ESI): m/z = 463, 461, 415, 315, 281 Elemental analysis: calculated for C_20_H_23_ClO_2_RuS_2_ C: 48.43%; H: 4.67%, found: C: 48.17%; H: 4.76%.

#### 3.2.6. [(η^6^-p-cymene)Ru(1-(3-hydroxyphenyl)-3-(ethylthio)-3-thioxo-prop-1-en-1-olate-O,S)Cl] (Ru5)

Synthesis was performed according to general procedure 1. [(η^6^-*p*-cymene)RuCl_2_]_2_ (500 mg, 0.81 mmol) was used. L5 (390 mg, 1.62 mmol) was dissolved in THF, *t*-BuOK (182 mg, 1.62 mmol) was added. Column chromatography mobile phase: DCM-DCM 10:THF 1-THF. Yield: 340 mg (41.0%) as red crystals. ^1^H NMR (600 MHz, CD_2_Cl_2_): δ = 1.26 (d, ^3^*J_H-H_* = 6.9 Hz, 6H, -cymene-CH-(C*H*_3_)_2_); 1.37 (t, ^3^*J_H-H_* = 7.5 Hz, 3H. –SCH_2_C*H*_3_); 2.21 (s, 3H, CH_3_, -cymene-C*H*_3_); 2.83 (m, 1H, -cymene-C*H*-(CH_3_)_2_); 3.23 (q, ^3^*J_H-H_* = 7.5 Hz, 2H, -SC*H*_2_CH_3_); 5.33 (m, 2H, -cymene:CH_3_-C-C*H*-CH-C-CH-(CH_3_)_2_); 5.52 (m, 2H –cymene:CH_3_-C-CH-C*H*-C-CH-(CH_3_)_2_); 6.68 (s, 1H, =C*H*); 6.83 (m, 2H, -Ar-*o*-H); 7.08 (m, 1H, -Ar-*m*-H); 7.20 (m, 3H, =C*H*/-Ar-*p*-H). ^13^C{^1^H} NMR (101 MHz, CD_2_Cl_2_): δ = 13.5 (-SCH_2_*C*H_3_); 18.0 (-cymene-C-*C*H_3_); 22.1 (-cymene-CH-(*C*H_3_)_2_); 25.6 (-S*C*H_2_CH_3_); 30.5 (-cymene-*C*H-(CH_3_)_2_); 82.9 (-cymene:CH_3_-C-*C*H-CH-C-CH-(CH_3_)_2_); 83.4 (CH_3_-C-*C*H-CH-C-CH-(CH_3_)_2_); 85.0 (CH_3_-C-CH-*C*H-C-CH-(CH_3_)_2_); 85.3 (CH_3_-C-CH-*C*H-C-CH-(CH_3_)_2_); 100.7 (CH_3_-*C*-CH-CH-C-CH-(CH_3_)_2_); 101.7 (CH_3_-C-CH-CH-*C*-CH-(CH_3_)_2_); 109.0 (=*C*H); 124.8 (-Ar-*m*-C); 129.0 (-*C*OH); 156.5 (-Ar-*p*-C); 177.9 (-Ar*C*1); 187.1 (-*C*-O-). MS (ESI): m/z = 518, 576, 474, 414, 328, 294, 292. Elemental analysis: calculated for C_21_H_25_ClO_2_RuS_2_ C: 49.45%; H: 4.94%, found: C: 49.29%; H: 5.02%.

#### 3.2.7. [(η^6^-p-cymene)Ru(1-(4-hydroxyphenyl)-3-(ethylthio)-3-thioxo-prop-1-en-1-olate-O,S)Cl] (Ru6)

Synthesis was performed according to general procedure 1. [(η^6^-*p*-cymene)RuCl_2_]_2_ (385 mg, 0.62 mmol) was used. L6 (300 mg, 1.25 mmol) was dissolved in THF, *t*-BuOK (140 mg, 1.25 mmol) was added. Column chromatography mobile phase: DCM-DCM 6:THF 1-THF. Yield: 100 mg (15.6%) as red crystals. ^1^H NMR (600 MHz, CD_2_Cl_2_): δ = 1.22 (d, ^3^*J_H-H_* = 6.9 Hz, 6H, -cymene-CH-(C*H*_3_)_2_); 1.33 (t, ^3^*J_H-H_* = 7.4 Hz, 3H, -SCH_2_C*H*_3_); 2.14 (s, 3H, -cymene-C*H*_3_); 2.79 (sp, *^3^J_H-H_* = 6.9, 1H, -cymene-C*H*-(CH_3_)_2_); 3.19 (q, ^3^*J_H-H_* = 7.4 Hz, 2H, -SC*H*_2_CH_3_); 5.23 (d, ^3^*J_H-H_* = 5.7 Hz, 2H, -cymene:CH_3_-C-C*H*-CH-C-CH-(CH_3_)_2_); 5.45 (d, ^3^*J_H-H_* = 17.4 Hz, 2H –cymene:CH_3_-C-CH-C*H*-C-CH-(CH_3_)_2_); 6.67 (s, 1H, =C*H*); 6.80 (d, 2H, ^3^*J_H-H_* = 8.7 Hz, -Ar-*o*-H); 7.68 (d, ^3^*J_H-H_* = 8.4 Hz, 2H, -Ar-*m*-H): 8.62 (s, 1H, -CO*H*). ^13^C{^1^H} NMR (101 MHz, CD_2_Cl_2_): δ = 13.9 (-SCH_2_*C*H_3_); 18.0 (-cymene-C-*C*H_3_); 22.3 (-cymene-CH-(*C*H_3_)_2_); 24.1(-S*C*H_2_CH_3_); 30.8 (-cymene-*C*H-(CH_3_)_2_); 83.0 (-cymene:CH_3_-C-*C*H-CH-C-CH-(CH_3_)_2_); 83.1 (CH_3_-C-*C*H-CH-C-CH-(CH_3_)_2_); 85.4 (CH_3_-C-CH-*C*H-C-CH-(CH_3_)_2_); 85.6 (CH_3_-C-CH-*C*H-C-CH-(CH_3_)_2_); 100.4 (CH_3_-*C*-CH-CH-C-CH-(CH_3_)_2_); 102.3 (CH_3_-C-CH-CH-*C*-CH-(CH_3_)_2_); 115.5 (=*C*H); 126.5 (-Ar-*o*-C); 129.2 (-Ar-C1); 131.5 (-Ar-*m*-C); 160.7 (-*C*OH); 177.8 (-*C*-O-); 185.0 (-*C*=S). MS (ESI): m/z = 476, 474, 414, 331, 301, 293. Elemental analysis: calculated for C_21_H_25_ClO_2_RuS_2_ C: 49.45%; H: 4.94%, found: C: 49.40%; H: 5.00%.

#### 3.2.8. [(η^6^-p-cymene)Ru(1-(2-methoxyphenyl)-3-(methylthio)-3-thioxo-prop-1-en-1-olate-O,S)] (Ru7)

Synthesis was performed according to general procedure 1. [(η^6^-*p*-cymene)RuCl_2_]_2_ (500 mg, 0.81 mmol) was used. L7 (389 mg, 1.62 mmol) was dissolved in THF, *t*-BuOK (182 mg, 1.62 mmol) was added. Column chromatography mobile phase: DCM-DCM 6:THF 1-THF. Yield: 700 mg (84.6%) as red crystals. ^1^H NMR (600 MHz, CD_2_Cl_2_): δ = 1.23 (d, ^3^*J_H-H_* = 6.9 Hz, 6H, -cymene-CH-(C*H*_3_)_2_); 2.19 (s, 3H,-cymene-C*H*_3_); 2.62 (s, 3H, -SC*H*_3_); 2.85 (sp, 1H, ^3^*J_H-H_* = 6.9 Hz, -cymene-C*H*-(CH_3_)_2_); 3.83 (s, 3H, -OC*H*_3_); 5.24 (d, ^3^*J_H-H_* = 6.1 Hz, 2H, -cymene:CH_3_-C-C*H*-CH-C-CH-(CH_3_)_2_); 5.46 (d, ^3^*J_H-H_* = 23.8 Hz, 2H –cymene:CH_3_-C-CH-C*H*-C-CH-(CH_3_)_2_); 6.64 (s, 1H, =C*H*); 6.92 (d, ^3^*J_H-H_* = 8.3 Hz, 1H, -Ar-*o*-H); 6.97 (m, 1H, -Ar-*m*-H); 7.38 (dd, ^3^*J_H-H_* = 7.7 Hz, ^4^*J_H-H_* = 1.8 Hz, 1H, -Ar-*p*-H); 7.50 (dd, ^3^*J_H-H_* = 7.6 Hz, ^4^*J_H-H_* = 1.7 Hz, 1H, -Ar-*m*-H). ^13^C{^1^H} NMR (101 MHz, CD_2_Cl_2_): δ = 17.5 (-SCH_3_); 18.1 (-cymene-C-*C*H_3_); 22.4/22.4 (-cymene-CH-(*C*H_3_)_2_); 30.8 (-cymene-*C*H-(CH_3_)_2_); 56.0 (-O*C*H_3_); 82.8 (-cymene:CH_3_-C-*C*H-CH-C-CH-(CH_3_)_2_); 83.3 (CH_3_-C-*C*H-CH-C-CH-(CH_3_)_2_); 85.4 (CH_3_-C-CH-*C*H-C-CH-(CH_3_)_2_); 100.7 (CH_3_-*C*-CH-CH-C-CH-(CH_3_)_2_); 102.6 (CH_3_-C-CH-CH-*C*-CH-(CH_3_)_2_); 111.9 (-Ar-*o*-C); 113.4 (=*C*H); 120.9 (-Ar-*m*-C); 129.2 (-Ar-C1); 130.6 (-Ar-*m*-C); 131.7 (-Ar-*p*-C); 156.9 (-Ar-*C*-OCH_3_); 179.0 (-*C*-O-); 185.9 (-*C*=S). MS (DEI): m/z = 503, 477, 475, 428, 341, 315, 281, 275. Elemental analysis: calculated for C_21_H_25_ClO_2_RuS_2_ C: 49.45%; H: 4.94%, found: C: 49.79%; H: 5.13%.

#### 3.2.9. [(η^6^-p-cymene)Ru(1-(3-methoxyphenyl)-3-(methylthio)-3-thioxo-prop-1-en-1-olate-O,S)] (Ru8)

Synthesis was performed according to general procedure 1. [(η^6^-*p*-cymene)RuCl_2_]_2_ (500 mg, 0.81 mmol) was used. L8 (389 mg, 1.62 mmol) was dissolved in THF, *t*-BuOK (182 mg, 1.62 mmol) was added. Column chromatography mobile phase: DCM-DCM 10:THF 1-THF. Yield: 450 mg (54.5%) as red crystals. ^1^H NMR (600 MHz, CDCl_3_): δ = 1.32 (d, ^3^*J_H-H_* = 6.9 Hz, 6H, -cymene-CH-(C*H*_3_)_2_); 2.25 (s, 3H, CH_3_, -cymene-C*H*_3_); 2.70 (s, 3H, -SC*H*_3_); 2.91 (sp, ^3^*J_H-H_* = 6.9 Hz, 1H, -cymene-C*H*-(CH_3_)_2_); 3.85 (s, 3H, -OC*H*_3_); 5.29 (d, ^3^*J_H-H_* = 24.0 Hz, 2H, -cymene:CH_3_-C-C*H*-CH-C-CH-(CH_3_)_2_); 5.46 (d, ^3^*J_H-H_* = 22.1 Hz, 2H –cymene:CH_3_-C-CH-C*H*-C-CH-(CH_3_)_2_); 6.76 (s, 1H, =C*H*); 6.99 (m, 1H, -Ar-*o*-H); 7.25–7.31 (m, 2H, -Ar-*m*-H); 7.34–7.40 (m, 2H, -Ar-*p*-H). ^13^C{^1^H} NMR (101 MHz, CD_2_Cl_2_): δ = 17.1 (-SCH_3_); 17.9 (-cymene-C-*C*H_3_); 22.5 (-cymene-CH-(*C*H_3_)_2_); 30.8 (-cymene-*C*H-(CH_3_)_2_); 55.1 (-O*C*H_3_); 82.7 (-cymene:CH_3_-C-*C*H-CH-C-CH-(CH_3_)_2_); 82.8 (CH_3_-C-*C*H-CH-C-CH-(CH_3_)_2_); 85.2 (CH_3_-C-CH-*C*H-C-CH-(CH_3_)_2_); 85.7 (CH_3_-C-CH-*C*H-C-CH-(CH_3_)_2_; 100.0 (CH_3_-*C*-CH-CH-C-CH-(CH_3_)_2_); 102.5 (CH_3_-C-CH-CH-*C*-CH-(CH_3_)_2_); 109.7 (=*C*H); 116.7 (-Ar-*o*-C); 120.0 (-Ar-*m*-C); 129.0 (-Ar-*p*-C); (-Ar-C1); 141.4 (-Ar-*m*-C);159.3 (-Ar-O*C*H_3_); 177.4 (-*C*-O-); 187.6 (-*C*=S). MS (ESI): m/z = 563, 474, 428. Elemental analysis: calculated for C_21_H_25_ClO_2_RuS_2_ C: 49.45%; H: 4.94%, found: C: 49.94%; H: 5.14%.

#### 3.2.10. [(η^6^-p-cymene)Ru(1-(4-methoxyphenyl)-3-(methylthio)-3-thioxo-prop-1-en-1-olate-O,S)] (Ru9)

Synthesis was performed according to general procedure 1. [(η^6^-*p*-cymene)RuCl_2_]_2_ (500 mg, 0.81 mmol) was used. L9 (389 mg, 1.62 mmol) was dissolved in THF, *t*-BuOK (182 mg, 1.62 mmol) was added. Column chromatography mobile phase: DCM-DCM 6:THF 1-DCM 4:THF 1-THF. Yield: 240 mg (29.1%) as red crystals. ^1^H NMR (600 MHz, CD_2_Cl_2_): δ = 1.53 (d, ^3^*J_H-H_* = 7.8 Hz, 6H, -cymene-CH-(C*H*_3_)_2_); 2.20 (s, 3H, CH_3_, -cymene-C*H*_3_); 2.67 (s, 3H, -SC*H*_3_); 2.85 (sp, ^3^*J_H-H_* = 7.8 Hz, 1H, -cymene-C*H*-(CH_3_)_2_); 3.85 (s, 3H, -OC*H*_3_); 5.27 (d, ^3^*J_H-H_* = 21.6 Hz, 2H, -cymene:CH_3_-C-C*H*-CH-C-CH-(CH_3_)_2_); 5.48 (d, ^3^*J_H-H_* = 18.7 Hz, 2H –cymene:CH_3_-C-CH-C*H*-C-CH-(CH_3_)_2_); 6.74 (s, 1H, =C*H*); 6.90 (d, ^3^*J_H-H_* = 8.8 Hz, 2H, -Ar-H); 7.81 (d, ^3^*J_H-H_* = 8.0 Hz, 2H, -Ar-H). ^13^C{^1^H} NMR (101 MHz, CD_2_Cl_2_): δ = 18.2 (-SCH_3_); 18.2 (-cymene-C-*C*H_3_); 22.4/22.4 (-cymene-CH-(*C*H_3_)_2_); 30.9 (-cymene-*C*H-(CH_3_)_2_); 55.8 (-O*C*H_3_); 83.1 (-cymene:CH_3_-C-*C*H-CH-C-CH-(CH_3_)_2_); 83.3 (CH_3_-C-*C*H-CH-C-CH-(CH_3_)_2_); 85.5 (CH_3_-C-CH-*C*H-C-CH-(CH_3_)_2_); 100.7 (CH_3_-*C*-CH-CH-C-CH-(CH_3_)_2_); 102.6 (CH_3_-C-CH-CH-*C*-CH-(CH_3_)_2_); 113.9 (-Ar-*o*-C); 113.4 (=*C*H); 120.9 (-Ar-*m*-C); 129.6 (-Ar-C1); 130.6 (-Ar-*m*-C); 131.7 (-Ar-*p*-C); 156.9 (-Ar-O*C*H_3_); 179.0 (-*C*-O-); 185.9 (-*C*=S). MS (ESI): m/z = 503, 477, 475, 429, 315, 281. Elemental analysis: calculated for C_21_H_25_ClO_2_RuS_2_ C: 49.45%; H: 4.94%, found: C: 49.60%; H: 5.08%.

#### 3.2.11. [(η^6^-p-cymene)Ru(1-(2-methoxyphenyl)-3-(ethylthio)-3-thioxo-prop-1-en-1-olate-O,S)] (Ru10)

Synthesis was performed according to general procedure 1. [(η^6^-*p*-cymene)RuCl_2_]_2_ (500 mg, 0.81 mmol) was used. L10 (389 mg, 1.62 mmol) was dissolved in THF, *t*-BuOK (182 mg, 1.62 mmol) was added. Column chromatography mobile phase: DCM-DCM 6:THF 1-THF. Yield: 700 mg (84.6%) as red crystals. ^1^H NMR (600 MHz, CD_2_Cl_2_): δ = 1.23 (d, ^3^*J_H-H_* = 6.9 Hz, 6H, -cymene-CH-(C*H*_3_)_2_); 1.38 (t, ^3^*J_H-H_* = 7.5 Hz, 3H, -SCH_2_C*H*_3_); 2.19 (s, 3H, -cymene-C*H*_3_); 2.84 (sp, ^3^*J_H-H_* = 6.9 Hz, 1H, -cymene-C*H*-(CH_3_)_2_); 3.20 (q, ^3^*J_H-H_* = 7.5 Hz, 2H, -SC*H*_2_CH_3_); 3.83 (s, 3H, -OC*H*_3_); 5.24 (d, ^3^*J_H-H_* = 6.0 Hz, 2H, -cymene:CH_3_-C-C*H*-CH-C-CH-(CH_3_)_2_); 5.45 (d, ^3^*J_H-H_* = 22.3 Hz, 2H –cymene:CH_3_-C-CH-C*H*-C-CH-(CH_3_)_2_); 6.63 (s, 1H, =C*H*); 6.92 (d, ^3^*J_H-H_* = 8.3 Hz, 1H, -Ar-*o*-H); 6.97 (t, 1H, -Ar-*m*-H); 7.38 (dd, ^3^*J_H-H_* = 7.8 Hz, ^4^*J_H-H_* = 1.8 Hz, 1H, -Ar-*p*-H); 7.51 (dd, ^3^*J_H-H_* = 7.6 Hz, ^4^*J_H-H_* = 1.8 Hz, 1H, -Ar-*m*-H). ^13^C{^1^H} NMR (101 MHz, CD_2_Cl_2_): δ = 13.9 (-SCH_2_C*H*_3_); 18.0 (-cymene-C-*C*H_3_); 22.3/22.4 (-cymene-CH-(*C*H_3_)_2_); 25.9 (-S*C*H_2_CH_3_); 30.8 (-cymene-*C*H-(CH_3_)_2_); 55.9 (-O*C*H_3_); 82.7 (-cymene:CH_3_-C-*C*H-CH-C-CH-(CH_3_)_2_); 83.2 (CH_3_-C-*C*H-CH-C-CH-(CH_3_)_2_); 85.4 (CH_3_-C-CH-*C*H-C-CH-(CH_3_)_2_); 85.5 (CH_3_-C-CH-*C*H-C-CH-(CH_3_)_2_); 100.8 (CH_3_-*C*-CH-CH-C-CH-(CH_3_)_2_); 102.5 (CH_3_-C-CH-CH-*C*-CH-(CH_3_)_2_); 111.9 (-Ar-*o*-C); 113.6 (=*C*H); 120.8 (-Ar-*m*-C); 129.2 (-Ar-C1); 130.6 (-Ar-*m*-C); 131.7 (-Ar-*p*-C); 156.9 (-Ar-*C*-OCH_3_); 179.2 (-*C*-O-); 185.0 (-*C*=S). MS (DEI): m/z = 502, 493, 489, 483, 428, 328, 296, 294. Elemental analysis: calculated for C_22_H_27_ClO_2_RuS_2_ C: 50.42%; H: 5.19%, found: C: 50.74%; H: 5.25%.

#### 3.2.12. [(η^6^-p-cymene)Ru(1-(3-methoxyphenyl)-3-(ethylthio)-3-thioxo-prop-1-en-1-olate-O,S)] (Ru11)

Synthesis was performed according to general procedure 1. [(η^6^-*p*-cymene)RuCl_2_]_2_ (500 mg, 0.81 mmol) was used. L11 (412 mg, 1.62 mmol) was dissolved in THF, *t*-BuOK (182 mg, 1.62 mmol) was added. Column chromatography mobile phase: DCM-DCM 10:THF 1-THF. Yield: 700 mg (82.4%) as red crystals. ^1^H NMR (600 MHz, CDCl_3_): δ = 1.32 (d, ^3^*J_H-H_* = 7.9 Hz, 6H, -cymene-CH-(C*H*_3_)_2_); 1.41 (t, ^3^*J_H-H_* = 7.4 Hz, 3H, -SCH_2_C*H*_3_); 2.91 (sp, ^3^*J_H-H_* = 7.9 Hz, 1H, -cymene-C*H*-(CH_3_)_2_); 3.01 (q, ^3^*J_H-H_* = 7.4 Hz, 2H, -SC*H*_2_CH_3_); 3.85 (s, 3H, -OC*H*_3_); 5.28 (d, ^3^*J_H-H_* = 22.7 Hz, 2H, -cymene:CH_3_-C-C*H*-CH-C-CH-(CH_3_)_2_); 5.51 (d, ^3^*J_H-H_* = 26.3 Hz, 2H –cymene:CH_3_-C-CH-C*H*-C-CH-(CH_3_)_2_); 6.74 (s, 1H, =C*H*); 6.99 (d, ^3^*J_H-H_* = 8.22 Hz, 1H, -Ar-*o*-H); 7.23-7.41 (m, 4H, -Ar-*m/p*-H). ^13^C{^1^H} NMR (101 MHz, CDCl_3_): δ = 13.7 (-SCH_2_C*H*_3_); 18.0 (-cymene-C-*C*H_3_); 22.5 (-cymene-CH-(*C*H_3_)_2_); 25.6 (-S*C*H_2_CH_3_); 30.5 (-cymene-*C*H-(CH_3_)_2_); 55.4 (-O*C*H_3_); 82.6 (-cymene:CH_3_-C-*C*H-CH-C-CH-(CH_3_)_2_); 82.8 (CH_3_-C-*C*H-CH-C-CH-(CH_3_)_2_); 85.1 (CH_3_-C-CH-*C*H-C-CH-(CH_3_)_2_); 85.6 (CH_3_-C-CH-*C*H-C-CH-(CH_3_)_2_; 100.1 (CH_3_-*C*-CH-CH-C-CH-(CH_3_)_2_); 102.6 (CH_3_-C-CH-CH-*C*-CH-(CH_3_)_2_); 109.7 (=*C*H); 116.8 (-Ar-*o*-C); 119.9 (-Ar-*m*-C); 126.3 (-Ar-*p*-C); 135.2 (-Ar-C1); 141.4 (-Ar-*m*-C);159.4 (-Ar-O*C*H_3_); 178.0 (-*C*-O-); 186.8 (-*C*=S). MS (ESI): m/z = 490, 488, 458, 428, 294. Elemental analysis: calculated for C_22_H_27_ClO_2_RuS_2_ C: 50.42%; H: 5.19%, found: C: 50.51%; H: 5.22%.

#### 3.2.13. [(η^6^-p-cymene)Ru(1-(4-methoxyphenyl)-3-(ethylthio)-3-thioxo-prop-1-en-1-olate-O,S)] (Ru12)

Synthesis was performed according to general procedure 1. [(η^6^-*p*-cymene)RuCl_2_]_2_ (500 mg, 0.81 mmol) was used. L12 (412 mg, 1.62 mmol) was dissolved in THF, *t*-BuOK (182 mg, 1.62 mmol) was added. Column chromatography mobile phase: DCM-DCM 6:THF 1-DCM 4:THF 1-THF. Yield: 790 mg (93.1%) as red crystals. ^1^H NMR (600 MHz, CDCl_3_): δ = 1.32 (d, ^3^*J_H-H_* = 7.8 Hz, 6H, -cymene-CH-(C*H*_3_)_2_); 1.41 (t, ^3^*J_H-H_* = 7.5 Hz, 3H, -SCH_2_C*H*_3_); 2.90 (sp, ^3^*J_H-H_* = 7.8 Hz, 1H, -cymene-C*H*-(CH_3_)_2_); 3.30 (q, ^3^*J_H-H_* = 7.5 Hz, 2H, -SC*H*_2_CH_3_); 3.86 (s, 3H, -OC*H*_3_); 2.85 (sp, 1H, -cymene-C*H*-(CH_3_)_2_); 3.85 (s, 3H, -OC*H*_3_); 5.29 (d, ^3^*J_H-H_* = 25.0 Hz, 2H, -cymene:CH_3_-C-C*H*-CH-C-CH-(CH_3_)_2_); 5.50 (d, ^3^*J_H-H_* = 22.5 Hz, 2H–cymene:CH_3_-C-CH-C*H*-C-CH-(CH_3_)_2_); 6.78 (s, 1H, =C*H*); 6.89 (d, ^3^*J_H-H_* = 8.8 Hz, 2H, -Ar-H); 7.80 (d, ^3^*J_H-H_* = 8.5 Hz, 2H, -Ar-H). ^13^C{^1^H} NMR (101 MHz, CDCl_3_): δ = 13.8 (-SCH_2_C*H*_3_); 18.1 (-cymene-C-*C*H_3_); 22.4 (-cymene-CH-(*C*H_3_)_2_); 25.6 (-S*C*H_2_CH_3_); 30.5 (-cymene-*C*H-(CH_3_)_2_); 55.4 (-O*C*H_3_); 82.9 (-cymene:CH_3_-C-*C*H-CH-C-CH-(CH_3_)_2_); 85.0 (CH_3_-C-*C*H-CH-C-CH-(CH_3_)_2_); 85.3 (CH_3_-C-CH-*C*H-C-CH-(CH_3_)_2_); 99.9 (CH_3_-*C*-CH-CH-C-CH-(CH_3_)_2_); 102.3 (CH_3_-C-CH-CH-*C*-CH-(CH_3_)_2_); 113.4 (-Ar-*o*-C); 113.4 (=*C*H); 126.3 (-Ar-*m*-C); 129.5 (-Ar-C1); 132.4 (-Ar-*m*-C); 132.4 (-Ar-*p*-C); 162.1 (-Ar-O*C*H_3_); 177.7 (-*C*-O-); 184.8 (-*C*=S). MS (ESI): m/z = 490, 488, 482, 428, 294. Elemental analysis: calculated for C_22_H_27_ClO_2_RuS_2_ C: 50.42%; H: 5.19%, found: C: 50.52%; H: 5.09%.

#### 3.2.14. [(η^6^-p-cymene)Ru(1-(2-ethoxyphenyl)-3-(methylthio)-3-thioxo-prop-1-en-1-olate-O,S)] (Ru13)

Synthesis was performed according to general procedure 1. [(η^6^-*p*-cymene)RuCl_2_]_2_ (500 mg, 0.81 mmol) was used. L13 (412 mg, 1.62 mmol) was dissolved in THF, *t*-BuOK (182 mg, 1.62 mmol) was added. Column chromatography mobile phase: DCM-DCM 10:THF 1-THF. Yield: 130 mg (23.7%) as red oil. ^1^H NMR (600 MHz, CDCl_3_): δ = 1.23 (d, ^3^*J_H-H_* = 7.1 Hz, 6H, -cymene-CH-(C*H*_3_)_2_); 1.35 (t, ^3^*J_H-H_* = 7.5 Hz, 3H, -OCH_2_C*H*_3_); 2.16 (s, 3H, CH_3_, -cymene-C*H*_3_); 2.57 (s, 3H, -SC*H*_3_); 2.82 (sp, ^3^*J_H-H_* = 7.1 Hz, 1H, -cymene-C*H*-(CH_3_)_2_); 4.29 (q, ^3^*J_H-H_* = 7.5 Hz, 2H, -OC*H*_2_CH_3_); 5.18 (d, ^3^*J_H-H_* = 16.0 Hz, 2H, -cymene:CH_3_-C-C*H*-CH-C-CH-(CH_3_)_2_); 5.39 (d, ^3^*J_H-H_* = 38.8 Hz, 2H –cymene:CH_3_-C-CH-C*H*-C-CH-(CH_3_)_2_); 6.76 (s, 1H, =C*H*); 6.78 (d, ^3^*J_H-H_* = 8.4 Hz, 1H, -Ar-*o*-H); 6.87 (m, 1H, -Ar-*m*-H); 7.25 (m, 1H, -Ar-*p*-H); 7.57 (dd, ^3^*J_H-H_* = 7.7 Hz, ^4^*J_H-H_* = 1.7 Hz, 1H, -Ar-*m*-H). ^13^C{^1^H} NMR (101 MHz, CDCl_3_): δ = 14.9 (-OCH_2_*C*H_3_); 17.9 (-SCH_3_); 17.9 (-cymene-C-*C*H_3_); 22.2 (-cymene-CH-(*C*H_3_)_2_); 30.4 (-cymene-*C*H-(CH_3_)_2_); 64.5 (-O*C*H_2_CH_3_); 82.6 (-cymene:CH_3_-C-*C*H-CH-C-CH-(CH_3_)_2_); 82.7 (CH_3_-C-*C*H-CH-C-CH-(CH_3_)_2_); 85.2 (CH_3_-C-CH-*C*H-C-CH-(CH_3_)_2_); 85.5 (CH_3_-C-CH-*C*H-C-CH-(CH_3_)_2_; 100.2 (CH_3_-*C*-CH-CH-C-CH-(CH_3_)_2_); 102.5 (CH_3_-C-CH-CH-*C*-CH-(CH_3_)_2_); 112.8 (=*C*H); 113.8 (-Ar-*o*-C); 117.2 (-Ar-*p*-C); 120.5 (-Ar-*o*-C); 130.6 (-Ar-*C*=C); 131.2 (-Ar-*m*-C); 131.4 (-Ar-C1); 155.9 (-*C*-O-); 177.7 (-*C*=S). MS (DEI): m/z = 524, 458, 119. Elemental analysis: calculated for C_22_H_27_ClO_2_RuS_2_ C: 50.42%; H: 5.19%, found: C: 50.39%; H: 5.32%.

#### 3.2.15. [(η^6^-p-cymene)Ru(1-(3-ethoxyphenyl)-3-(methylthio)-3-thioxo-prop-1-en-1-olate-O,S)] (Ru14)

Synthesis was performed according to general procedure 1. [(η^6^-*p*-cymene)RuCl_2_]_2_ (500 mg, 0.81 mmol) was used. L14 (412 mg, 1.62 mmol) was dissolved in THF, *t*-BuOK (182 mg, 1.62 mmol) was added. Column chromatography mobile phase: DCM-DCM 10:THF 1-THF. Yield: 471 mg (55.5%) as red crystals. ^1^H NMR (600 MHz, CDCl_3_): δ = 1.27 (d, ^3^*J_H-H_* = 10.3 Hz, 6H, -cymene-CH-(C*H*_3_)_2_); 1.40 (t, ^3^*J_H-H_* = 6.9 Hz, 3H, -OCH_2_C*H*_3_); 2.07 (s, 3H, CH_3_, -cymene-C*H*_3_); 2.66 (s, 3H, -SC*H*_3_); 2.86 (sp, ^3^*J_H-H_* = 10.3 Hz, 1H, -cymene-C*H*-(CH_3_)_2_); 4.03 (q, ^3^*J_H-H_* = 6.9 Hz, 2H, -OC*H*_2_CH_3_); 5.22 (d, ^3^*J_H-H_* = 8.8 Hz, 2H, -cymene:CH_3_-C-C*H*-CH-C-CH-(CH_3_)_2_); 5.43 (d, ^3^*J_H-H_* = 8.6 Hz, 2H –cymene:CH_3_-C-CH-C*H*-C-CH-(CH_3_)_2_); 6.71 (s, 1H, =C*H*); 6.94 (dd, ^3^*J_H-H_* = 12.18 Hz, ^4^*J_H-H_* = 3.0 Hz, 1H, -Ar-*p*-H); 7.22 (m, 1H, -Ar-*m*-H); 7.30 (s, 1H, -Ar-*o-*H); 7.33 (d, 1H, ^3^*J_H-H_* = 11.7 Hz, -Ar-*o-*H). ^13^C{^1^H} NMR (101 MHz, CDCl_3_): δ = 14.8 (-OCH_2_*C*H_3_); 17.2 (-SCH_3_); 17.9 (-cymene-C-*C*H_3_); 22.2 (-cymene-CH-(*C*H_3_)_2_); 30.5 (-cymene-*C*H-(CH_3_)_2_); 63.6 (-O*C*H_2_CH_3_); 82.6 (-cymene:CH_3_-C-*C*H-CH-C-CH-(CH_3_)_2_); 82.7 (CH_3_-C-*C*H-CH-C-CH-(CH_3_)_2_); 85.0 (CH_3_-C-CH-*C*H-C-CH-(CH_3_)_2_); 85.5 (CH_3_-C-CH-*C*H-C-CH-(CH_3_)_2_; 99.9 (CH_3_-*C*-CH-CH-C-CH-(CH_3_)_2_); 102.5 (CH_3_-C-CH-CH-*C*-CH-(CH_3_)_2_); 109.4 (=*C*H); 113.5 (-Ar-*o*-C); 117.2 (-Ar-*p*-C); 119.7 (-Ar-*o*-C); 128.9 (-Ar-*C*=C); 129.1 (-Ar-*m*-C); 141.3 (-Ar-C1); 158.7 (-*C*-O-); 187.4 (-*C*=S). MS (ESI): m/z = 488, 442, 314, 282. Elemental analysis: calculated for C_22_H_27_ClO_2_RuS_2_ C: 50.42%; H: 5.19%, found: C: 50.98%; H: 5.32%.

#### 3.2.16. [(η^6^-p-cymene)Ru(1-(4-ethoxyphenyl)-3-(methylthio)-3-thioxo-prop-1-en-1-olate-O,S)] (Ru15)

Synthesis was performed according to general procedure 1. [(η^6^-*p*-cymene)RuCl_2_]_2_ (500 mg, 0.81 mmol) was used. L15 (412 mg, 1.62 mmol) was dissolved in THF, *t*-BuOK (182 mg, 1.62 mmol) was added. Column chromatography mobile phase: DCM-DCM 10:THF 1-THF. Yield: 471 mg (55.5%) as red crystals. ^1^H NMR (600 MHz, CDCl_3_): δ = 1.25 (d, ^3^*J_H-H_* = 10.26 Hz, 6H, -cymene-CH-(C*H*_3_)_2_); 1.39 (t, ^3^*J_H-H_* = 7.0 Hz, 3H, -OCH_2_C*H*_3_); 2.19 (s, 3H, CH_3_, -cymene-C*H*_3_); 2.64 (s, 3H, -SC*H*_3_); 2.85 (sp, ^3^*J_H-H_* = 10.26 Hz, 1H, -cymene-C*H*-(CH_3_)_2_); 4.03 (q, ^3^*J_H-H_* = 7.0 Hz, 2H, -OC*H*_2_CH_3_); 5.21 (d, ^3^*J_H-H_* = 8.7 Hz, 2H, -cymene:CH_3_-C-C*H*-CH-C-CH-(CH_3_)_2_); 5.43 (d, ^3^*J_H-H_* = 8.7 Hz, 2H –cymene:CH_3_-C-CH-C*H*-C-CH-(CH_3_)_2_); 6.79 (s, 1H, =C*H*); 6.81 (d, ^3^*J_H-H_* = 13.38 Hz, 2H, -Ar-*m*-H); 7.76 (d, ^3^*J_H-H_* = 13.14 Hz, 2H, -Ar-*o*-H). ^13^C{^1^H} NMR (101 MHz, CDCl_3_): δ = 14.6 (-OCH_2_*C*H_3_); 17.1 (-SCH_3_); 17.9 (-cymene-C-*C*H_3_); 22.2 (-cymene-CH-(*C*H_3_)_2_); 30.4 (-cymene-*C*H-(CH_3_)_2_); 63.5 (-O*C*H_2_CH_3_); 82.7 (-cymene:CH_3_-C-*C*H-CH-C-CH-(CH_3_)_2_); 82.8 (CH_3_-C-*C*H-CH-C-CH-(CH_3_)_2_); 84.9 (CH_3_-C-CH-*C*H-C-CH-(CH_3_)_2_); 85.2 (CH_3_-C-CH-*C*H-C-CH-(CH_3_)_2_; 99.7 (CH_3_-*C*-CH-CH-C-CH-(CH_3_)_2_); 102.2 (CH_3_-C-CH-CH-*C*-CH-(CH_3_)_2_); 108.7 (=*C*H); 113.8 (-Ar-*m*-C); 128.9 (-Ar-*C*=C); 129.1 (-Ar-*o*-C); 132.1 (-Ar-C1); 161.4 (-Ar-*p*-C); 185.3 (-*C*=S). MS (ESI): m/z = 488, 442, 314, 282. Elemental analysis: calculated for C_22_H_27_ClO_2_RuS_2_ C: 50.42%; H: 5.19%, found: C: 50.67%; H: 5.32%.

#### 3.2.17. [(η^6^-p-cymene)Ru(1-(3-butoxyphenyl)-3-(methylthio)-3-thioxo-prop-1-en-1-olate-O,S)] (Ru16)

Synthesis was performed according to general procedure 1. [(η^6^-*p*-cymene)RuCl_2_]_2_ (500 mg, 0.81 mmol) was used. L14 (458 mg, 1.62 mmol) was dissolved in THF, *t*-BuOK (182 mg, 1.62 mmol) was added. Column chromatography mobile phase: DCM-DCM 10:THF 1-THF. Yield: 438 mg (49.0%) as red oil. ^1^H NMR (600 MHz, CDCl_3_): δ = 0.92 (t, 3H, -OCH_2_CH_2_CH_2_C*H*_3_); 1.28 (d, ^3^*J_H-H_* = 7.4 Hz, 6H, -cymene-CH-(C*H*_3_)_2_); 1.48 (m, 2H, -OCH_2_CH_2_C*H*_2_CH_3_); 1.76 (m, 2H, -OCH_2_C*H*_2_CH_2_CH_3_); 2.17 (s, 3H, CH_3_, -cymene-C*H*_3_); 2.62 (s, 3H, -SC*H*_3_); 2.87 (sp, ^3^*J_H-H_* = 7.4 Hz, 1H, -cymene-C*H*-(CH_3_)_2_); 3.98 (m, 2H, -OC*H*_2_CH_2_CH_2_CH_3_); 5.24 (d, ^3^*J_H-H_* = 5.7 Hz, 2H, -cymene:CH_3_-C-C*H*-CH-C-CH-(CH_3_)_2_); 5.43 (d, ^3^*J_H-H_* = 5.6 Hz, 2H –cymene:CH_3_-C-CH-C*H*-C-CH-(CH_3_)_2_); 6.72 (s, 1H, =C*H*); 6.95 (dd, ^3^*J_H-H_* = 8.16 Hz, ^4^*J_H-H_* = 2.2 Hz, 1H, -Ar-*p*-H); 7.23 (m, 1H, -Ar-*m*-H); 7.30 (s, 1H, -Ar-*o-*H); 7.33 (d, 1H, ^3^*J_H-H_* = 7.8 Hz, -Ar-*o-*H). ^13^C{^1^H} NMR (101 MHz, CDCl_3_): δ = 13.8 -OCH_2_CH_2_CH_2_*C*H_3_); 16.9 (-SCH_3_); 17.9 (-cymene-C-*C*H_3_); 19.8 (-OCH_2_CH_2_*C*H_2_CH_3_); 22.3 (-cymene-CH-(*C*H_3_)_2_); 30.5 (-cymene-*C*H-(CH_3_)_2_); 31.2 (-OCH_2_*C*H_2_CH_2_CH_3_); 67.8 (-O*C*H_2_CH_2_CH_2_CH_3_); 82.6 (-cymene:CH_3_-C-*C*H-CH-C-CH-(CH_3_)_2_); 82.7 (CH_3_-C-*C*H-CH-C-CH-(CH_3_)_2_); 85.1 (CH_3_-C-CH-*C*H-C-CH-(CH_3_)_2_); 85.5 (CH_3_-C-CH-*C*H-C-CH-(CH_3_)_2_; 99.8 (CH_3_-*C*-CH-CH-C-CH-(CH_3_)_2_); 102.6 (CH_3_-C-CH-CH-*C*-CH-(CH_3_)_2_); 109.4 (=*C*H); 113.5 (-Ar-*o*-C); 117.3 (-Ar-*p*-C); 119.0 (-Ar-*o*-C); 128.9 (-Ar-*C*=C); 128.9 (-*C*=C-); 129.0 (-Ar-*m*-C); 141.3 (-*C*1); 158.9 (-*C*-O-); 187.3 (-*C*=S). MS (ESI): m/z = 519, 516, 469, 315, 281, 278. Elemental analysis: calculated for C_24_H_31_ClO_2_RuS_2_ C: 52.21%; H: 5.66%, found: C: 52.60%; H: 5.62%.

#### 3.2.18. [(η^6^-p-cymene)Ru(1-(4-butoxyphenyl)-3-(methylthio)-3-thioxo-prop-1-en-1-olate-O,S)] (Ru17)

Synthesis was performed according to general procedure 1. [(η^6^-*p*-cymene)RuCl_2_]_2_ (500 mg, 0.81 mmol) was used. L17 (458 mg, 1.62 mmol) was dissolved in THF, *t*-BuOK (182 mg, 1.62 mmol) was added. Column chromatography mobile phase: DCM-DCM 10:THF 1-THF. Yield: 357 mg (40.0%) as red crystals. ^1^H NMR (600 MHz, CDCl_3_): δ = 0.95 (m, 3H, -OCH_2_CH_2_CH_2_C*H*_3_); 1.26 (d, ^3^*J_H-H_* = 7.0 Hz, 6H, -cymene-CH-(C*H*_3_)_2_); 1.46 (m, 2H, -OCH_2_CH_2_C*H*_2_CH_3_); 1.74 (m, 2H, -OCH_2_C*H*_2_CH_2_CH_3_); 2.19 (s, 3H, CH_3_, -cymene-C*H*_3_); 2.64 (s, 3H, -SC*H*_3_); 2.85 (sp, ^3^*J_H-H_* = 7.0 Hz, 1H, -cymene-C*H*-(CH_3_)_2_); 3.96 (m, 2H, -OC*H*_2_CH_2_CH_2_CH_3_); 5.25 (d, ^3^*J_H-H_* = 8.7 Hz, 2H, -cymene:CH_3_-C-C*H*-CH-C-CH-(CH_3_)_2_); 5.47 (d, ^3^*J_H-H_* = 8.7 Hz, 2H –cymene:CH_3_-C-CH-C*H*-C-CH-(CH_3_)_2_); 6.72 (s, 1H, =C*H*); 6.81 (d, ^3^*J_H-H_* = 8.8 Hz, 1H, -Ar-*m*-H); 7.76 (d, ^3^*J_H-H_* = 8.6 Hz, 2H, -Ar-*o*-H). ^13^C{^1^H} NMR (101 MHz, CDCl_3_): δ = 13.7 (-OCH_2_CH_2_CH_2_*C*H_3_); 17.1 (-SCH_3_); 17.9 (-cymene-C-*C*H_3_); 19.1 (-OCH_2_CH_2_*C*H_2_CH_3_); 22.3 (-cymene-CH-(*C*H_3_)_2_); 31.1 (-cymene-*C*H-(CH_3_)_2_); 33.6 (-OCH_2_*C*H_2_CH_2_CH_3_); 67.8 (-O*C*H_2_CH_2_CH_2_CH_3_); 82.7 (-cymene:CH_3_-C-*C*H-CH-C-CH-(CH_3_)_2_); 82.8 (CH_3_-C-*C*H-CH-C-CH-(CH_3_)_2_); 84.9 (CH_3_-C-CH-*C*H-C-CH-(CH_3_)_2_); 85.2 (CH_3_-C-CH-*C*H-C-CH-(CH_3_)_2_; 99.7 (CH_3_-*C*-CH-CH-C-CH-(CH_3_)_2_); 102.2 (CH_3_-C-CH-CH-*C*-CH-(CH_3_)_2_); 108.7 (=*C*H); 113.9 (-Ar-*o*-C); 128.9 (-Ar-*C*=C); 129.4 (-Ar-*o*-C); 132.0 (-Ar-*C*1); 161.6 (-*C*-O-); 185.2 (-*C*=S). MS (ESI): m/z = 519, 516, 469, 315, 281, 278. Elemental analysis: calculated for C_24_H_31_ClO_2_RuS_2_ C: 52.21%; H: 5.66%, found: C: 52.29%; H: 5.76%.

#### 3.2.19. [(η^6^-p-cymene)Os(1-(3-hydroxyphenyl)-3-(ethylthio)-3-thioxo-prop-1-en-1-olate-O,S)Cl] (Os3)

Synthesis was performed according to general procedure 1. [(η^6^-*p*-cymene)OsCl_2_]_2_ (500 mg, 0.63 mmol) was used. 3′-Hydroxy-β-hydroxydithiocin-namic acid methyl ester (286 mg, 1.26 mmol) was dissolved in THF, *t*-BuOK (140 mg, 1.26 mmol) was added. Column chromatography mobile phase: DCM-DCM 10:THF 1-THF. Yield: 520 mg (54.8%) as red crystals. ^1^H NMR (600 MHz, CDCl_3_): δ = 1.28 (d, ^3^*J_H-H_* = 6.7 Hz, 6H, -cymene-CH-(C*H*_3_)_2_); 2.31 (s, 3H, CH_3_, -cymene-C*H*_3_); 2.64 (s, 3H, -SC*H*_3_); 2.76 (sp, ^3^*J_H-H_* = 6.7 Hz, 1H, -cymene-C*H*-(CH_3_)_2_); 5.64 (s, 2H, -cymene:CH_3_-C-C*H*-CH-C-CH-(CH_3_)_2_); 5.82 (s, 2H –cymene:CH_3_-C-CH-C*H*-C-CH-(CH_3_)_2_); 6.88 (s, 1H, =C*H*); 6.91 (m, 1H, -Ar-*o*-H); 7.12 (t, 1H, -Ar-*m*-H); 7.26–7.28 (m, 2H, -Ar-*o*-H/-Ar-*p*-H). ^13^C{^1^H} NMR (101 MHz, CDCl_3_): δ = 17.5 (-SCH_3_); 18.1 (-cymene-C-*C*H_3_); 22.8 (-cymene-CH-(*C*H_3_)_2_); 30.8 (-cymene-*C*H-(CH_3_)_2_); 77.2 (-cymene:CH_3_-C-*C*H-CH-C-CH-(CH_3_)_2_); 77.4 (CH_3_-C-*C*H-CH-C-CH-(CH_3_)_2_); 92.9 (CH_3_-C-CH-*C*H-C-CH-(CH_3_)_2_); 93.2 (CH_3_-*C*-CH-CH-C-CH-(CH_3_)_2_); 110.7 (CH_3_-C-CH-CH-*C*-CH-(CH_3_)_2_); 114.4 (=*C*H); 118.3 (-Ar-*m*-C); 118.9 (-Ar-*o*-C); 129.2 (-*C*OH); 156.1 (-Ar-*p*-C); 174.7 (-Ar*C*1); 174.7 (-*C*-O-). MS (DEI): m/z = 586, 408. Elemental analysis: calculated for C_20_H_23_ClO_2_OsS_2_ C: 41.05%; H: 3.96%, found: C: 41.04%; H: 4.49%.

#### 3.2.20. [(η^6^-p-cymene)Os(1-(2-methoxyphenyl)-3-(methylthio)-3-thioxo-prop-1-en-1-olate-O,S)] (Os7)

Synthesis was performed according to general procedure 1. [(η^6^-*p*-cymene)OsCl_2_]_2_ (140 mg, 0.17 mmol) was used. 3′-Methoxy-β-hydroxydithiocin-namic acid methyl ester (85 mg, 0.35 mmol) was dissolved in THF, *t*-BuOK (39.7 mg, 0.35 mmol) was added. Column chromatography mobile phase: DCM-DCM 6:THF 1-THF. Yield: 80 mg (8.2%) as red crystals. ^1^H NMR (600 MHz, CDCl_3_): δ = 1.31 (d, ^3^*J_H-H_* = 7.0 Hz, 6H, -cymene-CH-(C*H*_3_)_2_); 2.31 (s, 3H,-cymene-C*H*_3_); 2.65 (s, 3H, -SC*H*_3_); 2.80 (sp, ^3^*J_H-H_* = 7.0 Hz, 1H, -cymene-C*H*-(CH_3_)_2_); 3.85 (s, 3H, -OC*H*_3_); 5.59 (d, ^3^*J_H-H_* = 5.1 Hz, 2H, -cymene:CH_3_-C-C*H*-CH-C-CH-(CH_3_)_2_); 5.80 (m, 2H –cymene:CH_3_-C-CH-C*H*-C-CH-(CH_3_)_2_); 6.87 (s, 1H, =C*H*); 7.02 (dd, ^3^*J_H-H_* = 8.2 Hz, ^4^*J_H-H_* = 1.9 Hz, 1H, -Ar-*o*-H); 6.97 (m, 1H, -Ar-*m*-H); 7.28 (m, 1H, -Ar-*p*-H); 7.35-7.40 (m, 1H, -Ar-*m*-H). ^13^C{^1^H} NMR (101 MHz, CDCl_3_): δ = 17.4 (-SCH_3_); 17.9 (-cymene-C-*C*H_3_); 22.7/22.9 (-cymene-CH-(*C*H_3_)_2_); 30.8 (-cymene-*C*H-(CH_3_)_2_); 55.4 (-O*C*H_3_); 73.9 (-cymene:CH_3_-C-*C*H-CH-C-CH-(CH_3_)_2_); 73.9 (CH_3_-C-*C*H-CH-C-CH-(CH_3_)_2_); 76.8 (CH_3_-C-CH-*C*H-C-CH-(CH_3_)_2_); 92.6 (CH_3_-*C*-CH-CH-C-CH-(CH_3_)_2_); 93.7 (CH_3_-C-CH-CH-*C*-CH-(CH_3_)_2_); 110.9 (-Ar-*o*-C); 112.7 (=*C*H); 116.6 (-Ar-*m*-C); 129.3 (-Ar-C1); 141.2 (-Ar-*m*-C); 159.5 (-Ar-*C*-OCH_3_); 174.9 (-*C*-O-); 186.7 (-*C*=S). MS (ESI): m/z = 565, 517, 371. Elemental analysis: calculated for C_21_H_24_ClO_2_OsS_2_ C: 42.09%; H: 4.21%, found: C: 42.75%; H: 4.14%.

#### 3.2.21. [(η^6^-p-cymene)Os(1-(2-ethoxyphenyl)-3-(methylthio)-3-thioxo-prop-1-en-1-olate-O,S)] (Os13)

Synthesis was performed according to general procedure 1. [(η^6^-*p*-cymene)OsCl_2_]_2_ (500 mg, 0.63 mmol) was used. 3′-Ethoxy-β-hydroxydithiocinnamic acid methyl ester (302 mg, 1.26 mmol) was dissolved in THF, *t*-BuOK (150 mg, 1.26 mmol) was added. Column chromatography mobile phase: DCM-DCM 10:THF 1-THF. Yield: 720 mg (72.4%) as red oil. ^1^H NMR (600 MHz, CDCl_3_): δ = 1.31 (d, ^3^*J_H-H_* = 7.0 Hz, 6H, -cymene-CH-(C*H*_3_)_2_); 1.44 (t, ^3^*J_H-H_* = 7.2 Hz, 3H, -OCH_2_C*H*_3_); 2.31 (s, 3H, CH_3_, -cymene-C*H*_3_); 2.66 (s, 3H, -SC*H*_3_); 2.80 (sp, ^3^*J_H-H_* = 7.0 Hz, 1H, -cymene-C*H*-(CH_3_)_2_); 4.08 (q, ^3^*J_H-H_* = 7.5 Hz, 2H, -OC*H*_2_CH_3_); 4.37 (t, ^3^*J_H-H_* = 7.2 Hz, 3H, OCH_2_C*H*_3_); 5.59 (d, ^3^*J_H-H_* = 15.0 Hz, 2H, -cymene:CH_3_-C-C*H*-CH-C-CH-(CH_3_)_2_); 5.80 (m, 2H, –cymene:CH_3_-C-CH-C*H*-C-CH-(CH_3_)_2_); 6.87 (s, 1H, =C*H*); 7.01 (dd, ^3^*J_H-H_* = 8.1 Hz, ^4^*J_H-H_* = 2.4 Hz, 1H, -Ar-*o*-H); 7.25-7.39 (m, 3H, -Ar-*m*-H/-Ar-*p*-H). ^13^C{^1^H} NMR (101 MHz, CDCl_3_): δ = 14.8 (-OCH_2_*C*H_3_); 17.4 (-SCH_3_); 17.9 (-cymene-C-*C*H_3_); 22.8 (-cymene-CH-(*C*H_3_)_2_); 30.8 (-cymene-*C*H-(CH_3_)_2_); 63.6 (-O*C*H_2_CH_3_); 73.8 (-cymene:CH_3_-C-*C*H-CH-C-CH-(CH_3_)_2_); 73.9 (CH_3_-C-*C*H-CH-C-CH-(CH_3_)_2_); 76.7 (CH_3_-C-CH-*C*H-C-CH-(CH_3_)_2_); 77.2 (CH_3_-C-CH-*C*H-C-CH-(CH_3_)_2_; 92.6 (CH_3_-*C*-CH-CH-C-CH-(CH_3_)_2_); 92.6 (CH_3_-C-CH-CH-*C*-CH-(CH_3_)_2_); 111.0 (=*C*H); 113.3 (-Ar-*o*-C); 117.2 (-Ar-*p*-C); 119.5 (-Ar-*o*-C); 129.2 (-Ar-*C*=C); 141.2 (-Ar-C1); 158.9 (-*C*-O-); 175.1 (-*C*=S). MS (EI): m/z = 614, 579. Elemental analysis: calculated for C_22_H_27_ClO_2_OsS_2_ C: 43.09%; H: 4.44%, found: C: 43.20%; H: 4.38%.

#### 3.2.22. [(η^6^-p-cymene)Os(1-(3-ethoxyphenyl)-3-(methylthio)-3-thioxo-prop-1-en-1-olate-O,S)] (Os14)

Synthesis was performed according to general procedure 1. [(η^6^-*p*-cymene)OsCl_2_]_2_ (500 mg, 0.63 mmol) was used. 4′-Ethoxy-β-hydroxydithiocinnamic acid methyl ester (302 mg, 1.26 mmol) was dissolved in THF, *t*-BuOK (150 mg, 1.26 mmol) was added. Column chromatography mobile phase: DCM-DCM 10:THF 1-THF. Yield: 160 mg (16.1%) as red crystals. ^1^H NMR (600 MHz, CDCl_3_): δ = 1.31 (d, ^3^*J_H-H_* = 6.5 Hz, 6H, -cymene-CH-(C*H*_3_)_2_); 1.57 (t, ^3^*J_H-H_* = 7.0 Hz, 3H, -OCH_2_C*H*_3_); 2.31 (s, 3H, CH_3_, -cymene-C*H*_3_); 2.58 (s, 3H, -SC*H*_3_); 2.80 (sp, ^3^*J_H-H_* = 6.5 Hz, 1H, -cymene-C*H*-(CH_3_)_2_); 4.10 (q, ^3^*J_H-H_* = 7.0 Hz, 2H, -OC*H*_2_CH_3_); 5.59 (d, ^3^*J_H-H_* = 17.1 Hz, 2H, -cymene:CH_3_-C-C*H*-CH-C-CH-(CH_3_)_2_); 5.79 (m, 2H –cymene:CH_3_-C-CH-C*H*-C-CH-(CH_3_)_2_); 6.85-6.95 (m, 3H, =C*H*/-Ar-*m*-H); 7.81-7.96 (m, 3H, -Ar-*p*-H/s, 1H, -Ar-*o-*H). ^13^C{^1^H} NMR (101 MHz, CDCl_3_): δ = 14.7 (-OCH_2_*C*H_3_); 17.9 (-SCH_3_); 22.8 (-cymene-CH-(*C*H_3_)_2_); 30.8 (-cymene-*C*H-(CH_3_)_2_); 63.6 (-O*C*H_2_CH_3_); 73.8 (-cymene:CH_3_-C-*C*H-CH-C-CH-(CH_3_)_2_); 74.0 (CH_3_-C-*C*H-CH-C-CH-(CH_3_)_2_); 76.7 (CH_3_-C-CH-*C*H-C-CH-(CH_3_)_2_); 77.0 (CH_3_-C-CH-*C*H-C-CH-(CH_3_)_2_; 114.1 (-Ar-*o*-C); 114.1 (-Ar-*p*-C); 129.3 (-Ar-*C*=C); 130.6 (-Ar-C1). MS (DEI): m/z = 614, 579. Elemental analysis: calculated for C_22_H_27_ClO_2_OsS_2_ C: 43.09%; H: 4.44%, found: C: 42.82%; H: 4.28%.

### 3.3. Structure Determination

The intensity data for the compounds were collected on a Nonius KappaCCD diffractometer using graphite-monochromated Mo-K_α_ radiation. Data were corrected for Lorentz and polarization effects; absorption was taken into account on a semi-empirical basis using multiple-scans [108,109]. The structures were solved by direct methods (SHELXS) and refined by full-matrix least squares techniques against Fo^2^ (SHELXL-97) [110]. All hydrogen atoms (with exception of the methyl-group at C13 of Ru14 and the methylene-group at C11 of L18 were located by difference Fourier synthesis and refined isotropically. All other hydrogen atoms were included at calculated positions with fixed thermal parameters. Crystallographic data as well as structure solution and refinement details are summarized in Table 4. MERCURY was used for structure representations [111].

### 3.4. Stability Determinations

NMR spectra were measured via NMR spectroscopy on Bruker Avance 400 MHz. Substances were solved in dmso-d_6_ or CD_2_Cl_2_ and measured directly at 37 °C or room temperature for 72 h. NS = 128 scans, t = 709 s/2891 s break, 72 measurements.

### 3.5. Biological Assays

Ovarian cancer cell lines were cultured under standard conditions (5% CO_2_, 37 °C, 90% humidity) in RPMI medium supplemented with 10% FCS, 100 U/mL penicillin and 100 µg/mL streptomycin (Life Technologies, Dreieich, Germany). Cisplatin (Sigma, Taufkirchen, Germany) was freshly dissolved at 1 mg/mL in 0.9% NaCl solution and diluted appropriately. New ruthenium(II) complexes and ligands were dissolved in dmso. Platinum-resistant A2780 and SKOV3 cells were established by repeated rounds of 3-day incubations with increasing amounts of Cisplatin starting with 0.1 µM. The concentration was doubled after 3 incubations, interrupted by recovery phases with normal medium. Cells that survived the third round of 12.8 µM Cisplatin were defined as resistant cultures. Determinations of IC50 values were carried out using the CellTiter96 non-radioactive proliferation assay (MTT assay, Promega, Mannheim, Germany). After seeding 5000 cells per well in a 96-well plate, cells were allowed to attach for 24 h and were incubated for 48 h with different concentrations of the substances ranging from 0 to 500 µM for Ruthenium and 0 to 1000 µM for ligand tests (0, 1, 10, 50, 100, 500, 1000 µm), for Cisplatin from 0 to 100 µM (0.1, 1, 5, 10, 50, 100 µM). Each measurement was done in triplicate and repeated 3 times. The proportion of viable cells was quantified by the MTT assay and after background subtraction relative values compared to the mean of medium controls were calculated. Non-linear regression analyses applying the Hill slope were run in GraphPad 5.0 software.

To examine cell cycle distribution and cell death rates, 30,000 cells were seeded in 12 well plates. After attaching for 24 h cells were treated with Cisplatin, Ru3 and Ru14 for 48 h at various concentrations for cell cycle and cell death analyses. For cell death analysis, immediately after treatment cells were stained with Propidium Iodid (PI) (1 µg/mL) on ice and the number of dead cells was measured using BD Canto II. For cell cycle distribution, cells recovered for 24 h after treatment. Afterwards, cells were fixed in ice-cold, 50% EtOH for 24 h at −20 °C. For DNA staining, fixed cells were incubated in PBS with 0.05% Triton-X, 0.1 µg/mL RNaseA and 50 µg/mL PI for 1 h at 4 °C in dark. DNA content was measured using BD Canto II.

For the determination of DNA damage induced by the treatment with different substances, histone γH2AX-foci were visualized by immunocytochemical staining. Cells were seeded on coverslips to reach 60–70% confluence after 24 h. After incubation (24 h) with different substances at IC_50_ concentrations for the resistant cells, cells were washed 3× with PBS and fixed for 10 min in 4% paraformaldehyde. Cells were again washed 3 times and then permeabilised by incubation with 0.25% Triton-X in PBS for 5min. Primary antibody against γH2AX (clone JBW301, Merck-Millipore, Darmstadt, Germany; diluted 1:2000) was incubated for 1 h at RT, and coverslips were washed 3 times afterwards. Alexa488-labelled secondary anti-mouse antibody (Life Technologies) was used in a 1:1000 dilution in PBS and applied for 1 h at RT. Cells were washed 3 times, counterstained with DAPI, washed again, and embedded in mounting medium (Vectorshield, Vector Laboratories, Burlingame, CA, USA). Slides were stored at 4 °C in darkness until microscopic evaluation was done using a Zeiss LSM 710 laser scanning microscope using a 63× oil-immersion objective. Image analysis was done using ImageJ and the FindFoci PlugIn [112].

## 4. Conclusions

In this work, we investigated 18 cinnamic acid derivatives, 17 ruthenium(II) complexes, and 4 osmium(II) complexes, and all of these compounds have been characterized by different methods, including X-ray diffraction analysis. NMR spectra signals have been compared to previously reported platinum(II) complexes and show significant changes in the ligand systems after complexation to metals. Stability determinations for some ruthenium(II) compounds were done with NMR spectroscopy, showing that these compounds are not stable in the solvent dmso, but in different other organic solvents. The biological activity of these complexes have been investigated mainly by IC50 measurements for all substances, as well as by cell cycle arrest, cell death, and DNA damage analyses for two of the ruthenium(II) complexes. Regarding the IC50 values, we can add to the previously reported SARs of ruthenium(II) and osmium(II) complexes by Keppler and coworkers that bearing an O,S-chelating ligand results in lower IC50 values for osmium(II) complexes compared to their ruthenium(II) analogues, but the ruthenium(II) compounds exhibit lower resistance factors [4]. Nevertheless, regarding non-cancerous cell lines, both complexes show a selective activity to cancer cell lines and high IC50 values on non-cancerous cells, pointing to possibly lower toxicity and side effects. The high cancer cell specific cytotoxic activity, also against cisplatin resistant cells combined with the diminished effects on cell cycle arrest and DNA damage point to a different mode of action. This may potentially involve the induction of ROS and mitochondrial dysfunction. Focusing on the structure-activity-relationship of the ruthenium(II) compounds, it is shown that longer alkyl chains at the aromatic ring lead to higher cytotoxic activity of these compounds. For the osmium complexes, most active compound is Os3, with a hydroxy-group at *meta*-position. Therefore, some of these compounds will be selected for further development, including in vivo experiments.

## Data Availability

The data presented in this study are available in the article and Appendix A. Crystallographic data (excluding structure factors) has been deposited with the Cambridge Crystallographic Data Centre as supplementary publication CCDC-1953506 for L14, CCDC-1953507 for L15, CCDC-1953508 for L17, CCDC-1953509 for L18, CCDC-1953503 for Ru9, CCDC-1953504 for Ru13, and CCDC-1953505 for Ru14. Copies of the data can be obtained free of charge on application to CCDC, 12 Union Road, Cambridge CB2 1EZ, UK (E-mail: deposit@ccdc.cam.ac.uk).

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
