# Peer review of "Highly Cytotoxic Osmium(II) Compounds and Their Ruthenium(II) Analogues Targeting Ovarian Carcinoma Cell Lines and Evading Cisplatin Resistance Mechanisms"

_ijms, 2022, doi:10.3390/ijms23094976_

Round 1

Reviewer 1 Report

The manuscript presents a really big study of anticancer properties of the series of Ru(II) and Os(II) compounds in comparison with well-known platinum(II) anticancer drugs. The manuscript itself is 33 pages long (7 of which are References). And there are also cif files and 16 pages of other supplementary. The introduction of the paper is, in fact, a mini-review of the field, which, in my opinion, is quite valuable in itself. The investigation of stability and speculations of the possible mechanisms of action are also very interesting.

Comments and suggestions:

  1. Abstract on page 1: “Structural characterizations and stability determinations have been carried out with standard techniques, including NMR spectroscopy and molecular structures” – there is no such standard technique as “molecular structures”.

  2. The review in the introduction is great, but while searching the literature on this topic, I found the paper on organo-osmium compounds which is also worth mentioning in it: https://pubs.rsc.org/en/content/articlelanding/2021/QI/C9QI01704F , especially since the authors mention the work of the same scientific group on ruthenium compounds.

  3. The quality of the pictures with NMR spectra is rather poor. For example, on Figure2.A on page 6 the top curve is barely visible. Also, the axis font size on Figure2.A is too small. It looks like the spectra images were somehow saved in raster format and then merged. It is better to export the spectra to some vector format or pdf and then use something like Inkscape to increase the fonts and thicken the lines. Or use Bruker's built-in tools for plots.

  4. The figure 3 on page 7 is not representative. It shows spectra tat t = 0, 24, 48 and 72 h after preparation of the sample. But the curves for t = 24, 48 and 72 h are identical. In the text it is stated that the measurements were done every hour, which can be seen in Supplementary. However, those pictures in supplementary are also a mess, since they contain too many curves. So, the Figure 3, in my opinion, should be replaced with something that really shows the changes in the spectra.

  5. In half of the article, the authors refer to cisplatin by its name, but on page 11 they introduce the abbreviation CDDP. This confuses. Either the abbreviation should be entered at the beginning of the article, or it should not be entered at all. Personally, I prefer the second option.

  6. On page 15: “The presented data suggest that Ru(II) and Os(II) complexes with O,S-chelating-Hydroxydithiocinnamic acid esters are both highly active and specific against cancer cell lines (Os(II) compounds) or Cisplatin resistant cells (Ru(II) compounds).” - are those cisplatin resistant cell lines not cancer?

Author Response

Response to Reviewer 1
The manuscript presents a really big study of anticancer properties of the series of Ru(II) and Os(II) compounds in comparison with well-known platinum(II) anticancer drugs. The manuscript itself is 33 pages long (7 of which are References). And there are also cif files and 16 pages of other supplementary. The introduction of the paper is, in fact, a mini-review of the field, which, in my opinion, is quite valuable in itself. The investigation of stability and speculations of the possible mechanisms of action are also very interesting.
Comments and suggestions:
1. Abstract on page 1: “Structural characterizations and stability determinations have been carried out with standard techniques, including NMR spectroscopy and molecular structures” – there is no such standard technique as “molecular structures”.
Authors reply: The reviewer is correct. We hanged the sentence and specifically stated the use of “…NMR spectroscopy and X-ray crystallography.” within the abstract
2. The review in the introduction is great, but while searching the literature on this topic, I found the paper on organo-osmium compounds which is also worth mentioning in it: https://pubs.rsc.org/en/content/articlelanding/2021/QI/C9QI01704F , especially since the authors mention the work of the same scientific group on ruthenium compounds.
Authors reply: Thank you for pointing us to this study. We included the study within the discussion (page 14: “Both, Ru(II) and Os(II) compounds can also inhibit proteosynthesis [34, 101].) because this paper points to a further potential mode of action that was missed within the first version.
3.  The quality of the pictures with NMR spectra is rather poor. For example, on Figure2.A on page 6 the top curve is barely visible. Also, the axis font size on Figure2.A is too small. It looks like the spectra images were somehow saved in raster format and then merged. It is better to export the spectra to some vector format or pdf and then use something like Inkscape to increase the fonts and thicken the lines. Or use Bruker's built-in tools for plots.
Authors reply: We have improved the quality of Figure 2 by enhancing the contrast and including new labels of the x-axis.
4. The figure 3 on page 7 is not representative. It shows spectra tat t = 0, 24, 48 and 72 h after preparation of the sample. But the curves for t = 24, 48 and 72 h are identical. In the text it is stated that the measurements were done every hour, which can be seen in Supplementary. However, those pictures in supplementary are also a mess, since they contain too many curves. So, the Figure 3, in my opinion, should be replaced with something that really shows the changes in the spectra.
Authors reply: The reviewer is correct that Figure 3 does not show the stepwise changes of the spectra. However, it clearly shows that the changes occur within 24 hours and we specifically state about the different time spans for specific changes within the text (page 7, first paragraph) and provide the full time course of spectra in the supplementary figure S2. We added a link to this figure in the legend of Fig. 3. Moreover, to improve the comprehensibility of Fig. S2 we added specific time points to the spectra.
 5. In half of the article, the authors refer to cisplatin by its name, but on page 11 they introduce the abbreviation CDDP. This confuses. Either the abbreviation should be entered at the beginning of the article, or it should not be entered at all. Personally, I prefer the second option.
Authors reply: Thank you for pointing to this inconsistency. We changed all statements of CDDP to cisplatin within the text, the figures and tables.
 6. On page 15: “The presented data suggest that Ru(II) and Os(II) complexes with O,S-chelating-Hydroxydithiocinnamic acid esters are both highly active and specific against cancer cell lines (Os(II) compounds) or Cisplatin resistant cells (Ru(II) compounds).” - are those cisplatin resistant cell lines not cancer?
Authors reply: We added the correct description as “cancer cell lines” also for the resistant subcultures.

Reviewer 2 Report

Some points to be considered:

  • The introduction needs to be shortened and includes more information about molecular docking studies in cancer.
  • The discussion should go into further depth, discussing your current findings and concluding with future directions, as well as the advantages of this work in cancer patients' medication treatments.
  • A validation of the docking procedure is mandatory for all of the studied compounds.
  • The rational of the work is more clear now but authors should clarify better the aim of the work with this scaffold.
  • Manuscript should be checked for clerical errors.

Author Response

Response to Reviewer 2
Some points to be considered:
1.    The introduction needs to be shortened and includes more information about molecular docking studies in cancer. 
A validation of the docking procedure is mandatory for all of the studied compounds.
Authors reply: Thank you for pointing to the use of molecular docking studies. However, because we don´t know the specific target of the new compounds, yet, we see no possibility to run such studies. However, we included a statement within the discussion for future analyses (page 14: “A direct interaction of Ru3 with a model protein (RNaseA) resulted in ligand exchange, binding to histidine residues and altered coordination sphere geometry, pointing to a mode-of-action that involves protein targets [50]. Future studies may identify the specific target proteins enabling molecular docking studies and specific refinement of the organo-metal compound structure.”) 
2.    The discussion should go into further depth, discussing your current findings and concluding with future directions, as well as the advantages of this work in cancer patients' medication treatments.
Authors reply: Although we understand the importance to discuss the future influence of medication treatments we think more experiments (animal models, 3D cell culture) are needed as stated within the manuscript before an in-depth discussion is possible. Thus we did not include such a discussion in the revised version. However, we included a short statement at the end of the results and discussion section (page 15: “If future studies can solve these limitations and validate the high cancer cell specific cytotoxicity also against platin resistant tumors these compounds are likely to improve the treatment of ovarian cancer patients.).
3. The rational of the work is more clear now but authors should clarify better the aim of the work with this scaffold.
Authors reply: We added a specific statement about the aim in the abstract (“Aim of this study was to determine the anticancer activity and the ability to evade platin resistance mechanisms for these compounds.”)
 4. Manuscript should be checked for clerical errors.
Authors reply: We checked the manuscript.